# Opportunities for decentralised solar power to improve reliability, reduce emissions and avoid stranded assets

Philip Sandwell [1,2,6], Benedict Winchester [1,2,3,6], Shivika Mittal [2,4], Christos N. Markides [3], Hamish Beath [1,2,5] ✉ & Jenny Nelson [1,2] ✉

Despite recent improvements to electricity access in lower-income countries, reliability remains low for many. Local renewable electricity infrastructure supplementing the national grid offers a promising route to improved reliability for rural communities. However, improvements in the reliability of national grids create risks for investors including the possibility of "stranded" renewable assets. We use energy-system modelling to explore ways in which solar photovoltaic (PV)-based mini-grids could be interconnected with national grids. We explore the impact of reduced electricity demand to quantify the investment risks of losing customers. Our results indicate that national grid–connected mini-grids can reduce the unit electricity costs for communities whilst also increasing reliability and reducing the carbon intensity of electricity in line with Sustainable Development Goal (SDG) 7. Reductions in demand have a minimal impact at lower levels but at moderate levels are likely to undermine economic viability. Finally, we discuss policy interventions to facilitate and protect investing in national grid-connected mini-grids.

The risk of "stranded" assets—defined as having undergone premature or unanticipated write-downs—is often considered for fossil-fuel energy infrastructure in the face of climate action and policy[1–3]. However, the risk of stranding also exists in some contexts for low-carbon energy infrastructure but has not been widely discussed[4,5]. Investment in decentralised renewables-based off-grid solutions for energy access, such as solar photovoltaic (PV) mini-grids, has been inhibited by the risk of the arrival of national grid infrastructure as governments enact electricity-access plans[5–7]. Electricity tariffs from the national grid in low-income countries[8] like India and Ghana are significantly lower than those charged by mini-grids at cost recovery levels[9]. This is often due to subsidies or economies of scale. This could therefore compromise the ability of the decentralised infrastructure to provide a return on investment if competing side-by-side after the arrival of the national

grid[10–12]. There is also a risk that mini-grid assets may be taken over by the government or distribution companies once the main grid arrives, potentially without proper compensation[7]. Consequently, there is a risk that the solar and battery assets installed to supply customers on the mini-grid may become stranded. Without clear policy support, such as designating zones for off-grid solutions, national grid-compatible technical standards, and commercial arrangements for generation and distribution assets[6], these risks limit the appetite of investors and companies for developing low-carbon mini-grid electricity infrastructure.

Mini-grids have the potential to offer consumers more reliable access to electricity compared to national grids[13,14], provided they are well-maintained, and protected from severe weather[15]. Additionally, depending on factors such as population density, demand levels, and

[1]Department of Physics, Imperial College London, London, UK. [2]Grantham Institute—Climate Change and the Environment, Imperial College London, London, UK. [3]Clean Energy Processes (CEP) Laboratory, Department of Chemical Engineering, Imperial College London, London, UK. [4]CICERO Center for International Climate Research, Oslo, Norway. [5]Centre for Environmental Policy, Imperial College London, London, UK. [6]These authors contributed equally: Philip Sandwell, Benedict Winchester. ✉e-mail: hamish.beath16@imperial.ac.uk; jenny.nelson@imperial.ac.uk

proximity to existing distribution infrastructure, mini-grids can connect communities at lower lifetime costs than national grid expansion[16]. Reliable electricity helps to ensure consumers can see the full benefits from their connection[17], with advantages such as improved household incomes[18]. Integrating mini-grid infrastructure into national networks can allow local renewable assets to help increase service hours where the national grid is intermittent[19–21]. Current policies in countries with electricity access challenges are either ambiguous or prohibit such integration[22–24]. Several studies have highlighted the uncertain regulatory environment as one of the challenges to scaling mini-grids in low-income countries[6,25–27]. Changes to national grid infrastructure would be necessary to accommodate these interconnections[20]. Still, this synergistic relationship could avoid infrastructure becoming redundant whilst strengthening the quality of supply[21] and could operate under a range of commercial models and policy frameworks[28].

India is a country where these considerations are salient. Although India has achieved remarkable progress toward universal electricity access[22,29], the picture of rural electricity supply remains complex, with the proportion of households with a connection varying significantly across states[30]. Despite substantial progress, made possible by the extension of the national grid network, the quality of supply remains a critical issue in rural areas[10,22,31] where customers experience unpredictability in terms of the timing, duration, and availability of national grid electricity[28,31,32]. Reliability data at a high spatial resolution for the rural electricity grid in India is sparse. Still, a recent study estimated that households in Uttar Pradesh, a populous and predominantly rural state in northern India, had access to only 8.1 hours of grid electricity per day on average in 2019, an improvement of just 0.6 hours compared to 2014[33]. More recent official figures suggest that grid reliability in rural areas in Uttar Pradesh is much higher, but has fallen each year between 2019 and 2021[34]. To meet supply shortfalls, the expansion of both utility-scale and decentralised solar PV features heavily in state and national energy strategies[35–37]. This will help contribute to India's updated Nationally Determined Contribution (NDC) to reduce the greenhouse gas (GHG) emissions intensity of its gross domestic product (GDP) by 45% compared to 2005 levels by 2030. This would translate to a reduction of the emissions intensity of grid electricity from around 600 $gCO_2eq\,kWh^{-1}$ at present[38,39] to around 444 $gCO_2eq\,kWh^{-1}$ in 2030[40,41]. Mini-grid renewable energy infrastructure in rural areas could provide improved reliability and help meet national climate mitigation and sustainable development goals simultaneously[42].

Greater reliability increases the prices households in India are willing to pay for electricity and also increases the revenues that rural enterprises make, thereby increasing the amount they can afford to pay for electricity[43–47]. The willingness to pay for better reliability is evidenced by Indian mini-grid companies charging tariffs of 0.22–0.55 United-States Dollars ($) $kWh^{-1}$: these rates surpass the national grid tariffs, highlighting the value consumers place on improved service[46,48,49]. Household satisfaction with electricity services in India is strongly linked to grid reliability: increasing the supply duration by one standard deviation can have the same effect on consumer satisfaction as providing electricity for the first time[50]. This suggests the higher electricity prices with mini-grids may not be a significant issue if the quality of service is superior to the national grid.

To explore the potential for mini-grids to improve service reliability and affordability, we need to assess alternative solutions for providing reliable, sustainable and affordable electricity to rural communities whilst national grids remain unreliable in many countries[51]. Energy system modelling can allow us to evaluate and quantify the potential for systems composed of grid-connected renewable mini-grid infrastructure to provide low-cost and -carbon power. Previous studies have used a single reliability level for the national grid network[52–55]. However, it is important to explore the

impacts at different national grid reliability levels given that it may differ from what is anticipated, or change over time.

We employ an open-source modelling framework[56,57] to assess the potential for national grid-connected solar PV mini-grids to improve hours of electricity supply service for consumers (Fig. 1). We consider how integrating mini-grid infrastructure at different levels of reliability may impact the risk of stranded assets and the potential loss of asset value under various scenarios. We also explore the risk of stranded assets in the event of demand loss from loss of customers if the national grid improves in reliability. We quantify the costs and GHG emissions to evaluate their long-term financial and environmental sustainability. We consider a location in Uttar Pradesh, India, as an example region in which rural electricity access has been well explored via primary surveying[32,45,46,58]. However, our findings are relevant for other contexts beyond India where direct competition between national grids and mini-grids may occur. Most notably, significant electricity access challenges persist in sub-Saharan Africa, including in areas of relatively high population density and close to local grid infrastructure[59]. Globally, it is estimated that 3.5 billion people live with an unreliable electricity connection: local mini-grid infrastructure could alleviate these challenges in some contexts[60].

The community that we represent in the model comprises 547 households and 66 enterprises[32] with an average daily load of 761 kWh, dominated by domestic energy (695 kWh day$^{-1}$ on average) and the average national grid availability at the time of data collection was 12.5 h of service per day[32] (see "Methods", Supplementary Note 1 and Supplementary Fig. 1). We optimise electricity systems supplied by solar PV generation with battery storage and the national grid. We define the optimum system as meeting the required minimum service hours at the lowest levelised cost of used electricity (LCUE) (see "Methods").

Here, we find that by using grid-connected mini-grids, reliability can be improved whilst also seeing benefits relating to the cost and emissions intensity of electricity. When the national grid availability is below 16 h, there is a reduction in the unit cost of supply of electricity seen when the mini-grid infrastructure provides 4 or fewer hours of service. In most cases, except for the highest national grid reliability cases, adding mini-grid infrastructure reduces the emissions intensity of electricity. Our results demonstrate that investments are likely to not be stranded by moderate improvements to national-grid reliability, although at higher levels stranding risk is high. Finally, we demonstrate that the loss of consumers on the network (lower electricity demand) may create stranding risk, but only at higher levels.

## Results
### Optimisation results for varying levels of grid and service availability

Installing mini-grid assets in a region where national grid electricity is already established can provide reliability benefits and increase the potential revenue that can be generated from selling electricity. Relevant results are shown in Fig. 2. The higher the required service hours, the greater the number of local generation and storage assets that must be installed (Fig. 2a, b), and so the up-front costs for developers, and the investment risk, are also higher.

The effect of an increase in local mini-grid infrastructure is reflected in the overall system LCUE (Fig. 2c). Systems that significantly improve minimum service hours above the national grid availability incur the highest LCUEs of mini-grid systems with local renewable assets installed: mini-grids providing 2 h of service beyond the national grid have LCUEs of 0.19–0.24 $ $kWh^{-1}$ compared to LCUEs of 0.23–0.28 $ $kWh^{-1}$ for eight additional service hours (Fig. 2c). The increase in LCUE for providing additional service hours compared to national grid-only systems is discussed in detail below.

It should be noted that these values for the LCUE refer to the costs incurred per levelised cost of providing electricity which is consumed.

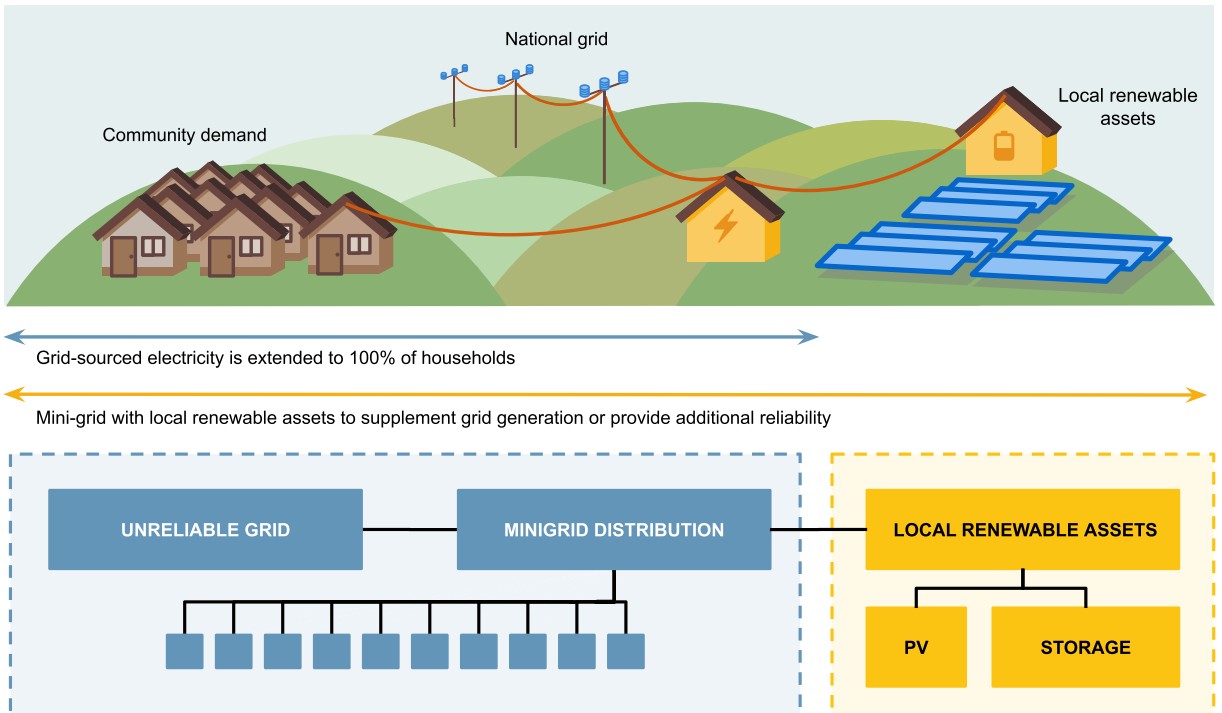

**Fig. 1 | Illustration showing the energy access scenarios considered.** Additional mini-grid distribution infrastructure has been installed in the case-study community to extend access to all households with power either coming from just a connection to the national-grid network or from both local renewable assets and a national-grid connection. Local renewable assets within the mini-grid system provide electricity above national grid availability, enhancing the total service hours.

Not included in this metric is the actual cost which would be charged to a consumer, where mini-grid companies would likely seek to recover costs incurred and generate a profit over the lifetime of the system. Rather than incorporate an estimate of likely profit margins, we here use the widely utilised metric of LCUE to enable a good comparison between our systems and alternative means of supplying electricity.

There are also systems (Fig. 2c) where electricity access is extended to the entire community through expanded distribution infrastructure, but where no local renewable assets are installed. These systems, for which the national grid availability exceeded the requirements of the system, also result in LCUE values at the higher end of the systems considered. The LCUE values for these systems are higher than or equal to those for systems where local renewable assets are installed for the same hours of grid availability other than systems that supply upwards of 22 h of electricity per day. We attribute this to the levelised costs of the additional distribution infrastructure being lower when renewable assets are installed as more electricity is supplied by these assets at a lower cost.

For the majority of optimum PV and storage mini-grid systems, the average emissions intensity of the electricity used (Fig. 2d) is below India's NDC goal (531 gCO$_2$eq kWh$^{-1}$ on average; although we assume a reduction over time due to increased renewables penetration, see "Methods")[3,41,61], where only systems which are sized for high levels of service (20 or more hours per day) when the national-grid network is readily available (for 18 or more hours per day on average) are more carbon intensive. National grid-only systems (i.e., connection and distribution infrastructure supplying only grid electricity) have emissions intensities of 599 to 607 gCO$_2$eq kWh$^{-1}$, far above the NDC target of 531 gCO$_2$eq kWh$^{-1}$.

**Costs of providing service beyond the national grid**
Installing local electricity generating and storage mini-grid assets can provide both additional service hours beyond those provided by the national grid and a reduction in LCUE compared to relying solely on national grid power (Figs. 2c and 3a). This occurs when both the additional hours provided and the national grid availability are below a certain threshold, around 16 h, and the local mini-grid assets can provide a minimum of 4 additional service hours (Fig. 3a). In these instances, the reduction in the LCUE results from the discounted energy supplied increasing faster than total costs (see "Methods"). The values for the LCUE premium are compared to installed connection and distribution infrastructure supplying only electricity sourced from the national grid. In other words, when modelling scenarios with only national grid electricity, we include the cost of localised connection and distribution infrastructure. A sensitivity analysis whereby the subsidy for grid electricity was removed is shown in Supplementary Fig. 3 (see Supplementary Note 3).

Our results indicate that a PV-and-storage mini-grid, installed with a national grid connection, can reduce the electricity costs for the community whilst also bringing benefits in terms of reliability and meeting energy-access targets in line with Sustainable Development Goal (SDG) 7. There is a noticeable trend that shows a lower increase in the LCUE premium as the number of additional service hours provided decreases (Fig. 3a). Our results also show that providing additional service hours beyond the national grid through the use of mini-grid assets can decrease the emissions intensity of electricity used in almost all cases except when 1 h of additional service is provided while the national grid is available for 23 h per day (Fig. 3b). These two trends, taken together, demonstrate that providing additional service hours can, in some cases, reduce both the levelised costs of energy consumed and the associated GHG emissions produced.

**Avoiding stranded assets as national grid reliability improves**
The availability of the national grid network depends on location[32,62] and may change over time. This could lead to local renewable assets being underutilised or insufficient in the years following their installation. As a result, there is also a risk of value loss of local assets if the national grid availability is different from mini-grid developer

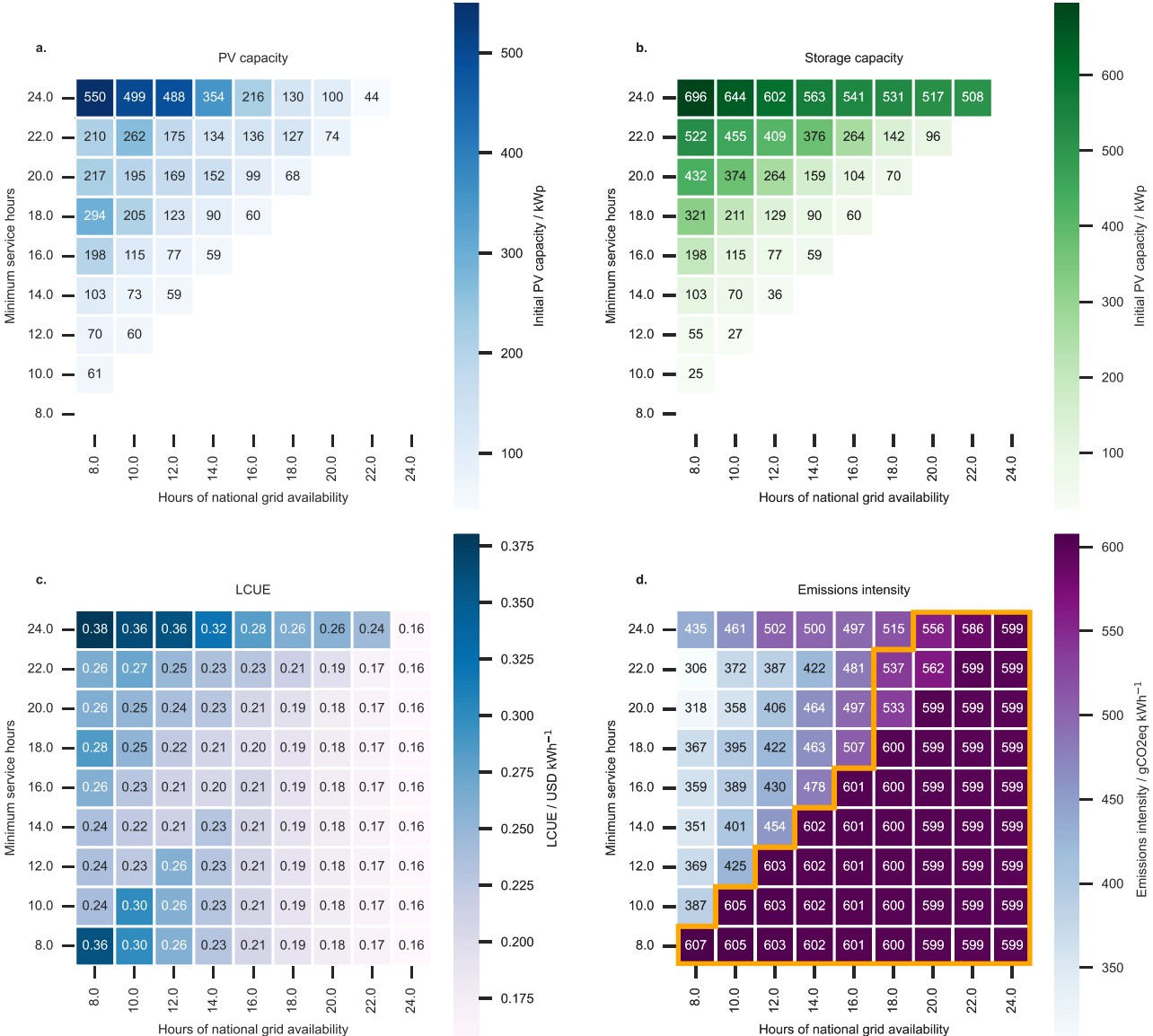

**Fig. 2 | Optimum systems that provide a desired level of service availability (vertical axes) for a given grid availability (horizontal axes).** The photovoltaic (PV) (**a**) and storage (**b**) capacities are shown which yield the lowest levelised cost of used electricity (LCUE) (**c**). The average emissions intensity of all electricity used (**d**) is shown relative to the Government of India's target for 2030 (531 gCO₂eq kWh⁻¹ on average[41] between 2022 and 2030). Systems that exceed the 2030 target are outlined in yellow. Diesel generators are not considered. Where the grid availability exceeds the minimum service hours, only the grid is used, exceeding the service-availability requirement if possible. The non-monotonic nature of the LCUE (**c**) and PV system capacities (**a**) as the minimum service hours increase is due to increases in the storage capacity (**b**), needed to meet additional night-time hours of service.

expectations. Depending on whether a developer can continue to bring in revenues, this may put these assets at risk of stranding. To investigate this, we consider grid-connected mini-grid systems sized for our baseline national grid-availability profile (13 h of service on average per day) but consider the implications of higher or lower national grid availability. We investigate both increases and decreases in the daily average number of service hours provided by the national grid network.

Figure 4 shows the impact on the LCUE for mini-grid systems sized for 13 h of national grid availability to meet a given level of service hours when the reliability of the national grid increases (to the right) or decreases (to the left). Highlighted in pink are systems for which a change in the reliability of the national grid results in an increase in the overall LCUE. We highlight those assets in the top-right which experience an increase in the LCUE as being at risk of "asset stranding." For these systems, an increase in the reliability of the national grid results

in electricity that was provided by the installed PV and battery infrastructure being supplanted by that sourced from the grid. This means that the installed assets are oversized. However, the maximum excess LCUE resulting from this oversizing is 0.11 $ kWh⁻¹, which falls within the range of tariffs charged by mini-grid companies in India[46,48,49]. Since the increase in the LCUE is of a similar order of magnitude to existing tariffs, mini-grid companies may be able to accommodate this additional cost by charging customers a slightly higher rate.

Similarly, for systems in the bottom-left (originally sized for 14–17 h of service), a decrease in national grid availability can result in "undersized assets" (Fig. 4). This happens at critical times, when there is not enough supply from the national grid and the mini-grid assets are unable to meet the demand for which the system originally relied on the national grid. This means the mini-grid assets no longer meet sufficient demand which in turn results in the increase in the LCUE. The point at which these assets become undersized is influenced by

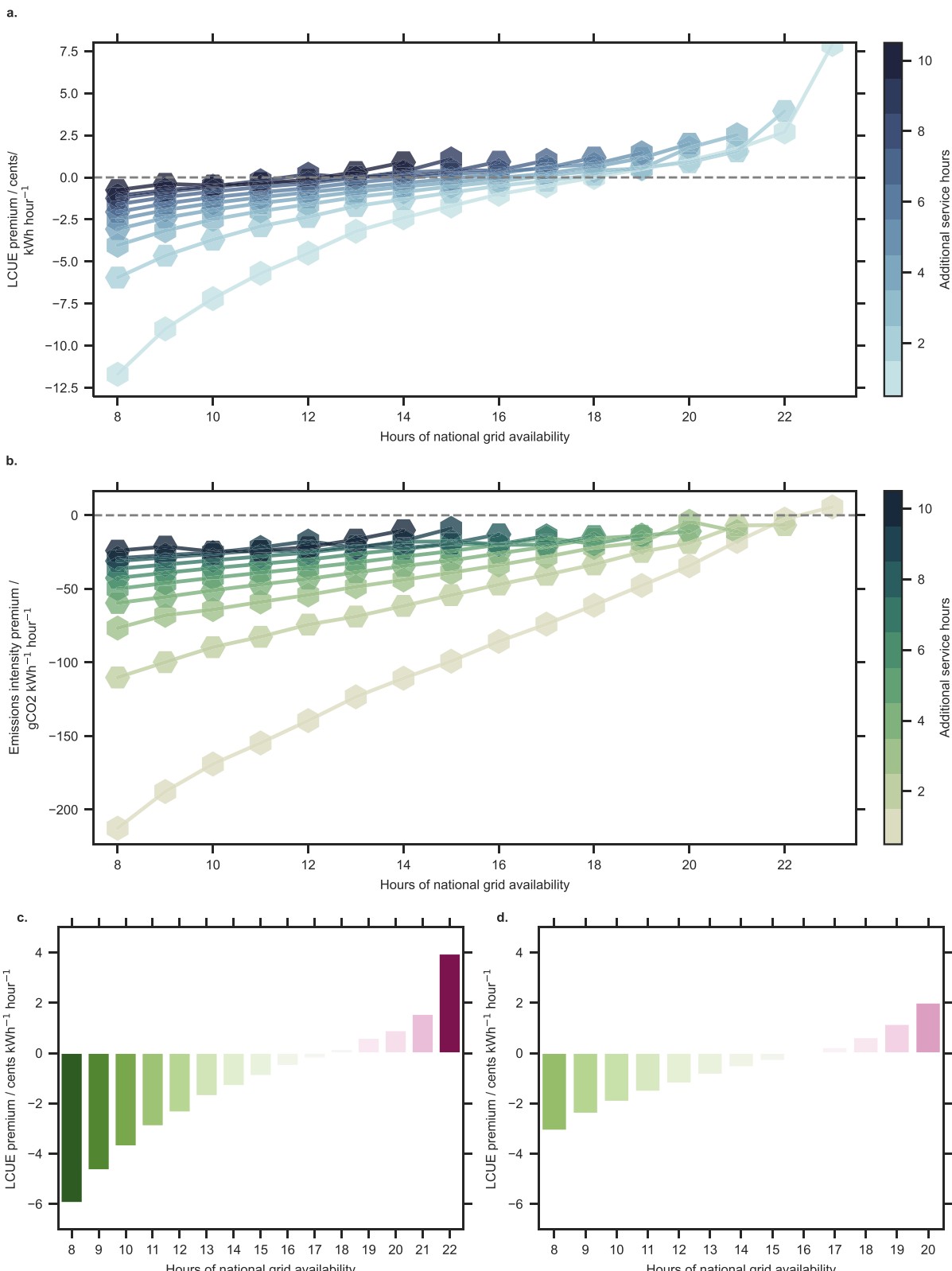

**Fig. 3 | LCUE and emissions intensity premiums for additional service hours beyond those provided by the grid.** The values plotted represent **a**, **c** and **d** the additional cost, levelised per unit of electricity, per hour of additional service required beyond that which the national grid can provide and **b** the additional emissions intensity per hour of additional service required. Connected points (of the same colour) show systems of equal additional service provision beyond the grid as the availability of grid electricity is varied. The change in **a** the LCUE and **b** the emissions intensity are shown for values of national grid availability between 8 and 23 h for 1–10 h of additional service. The change in LCUE is shown for an additional 2 h of service (**c**) and an additional 4 h of service (**d**).

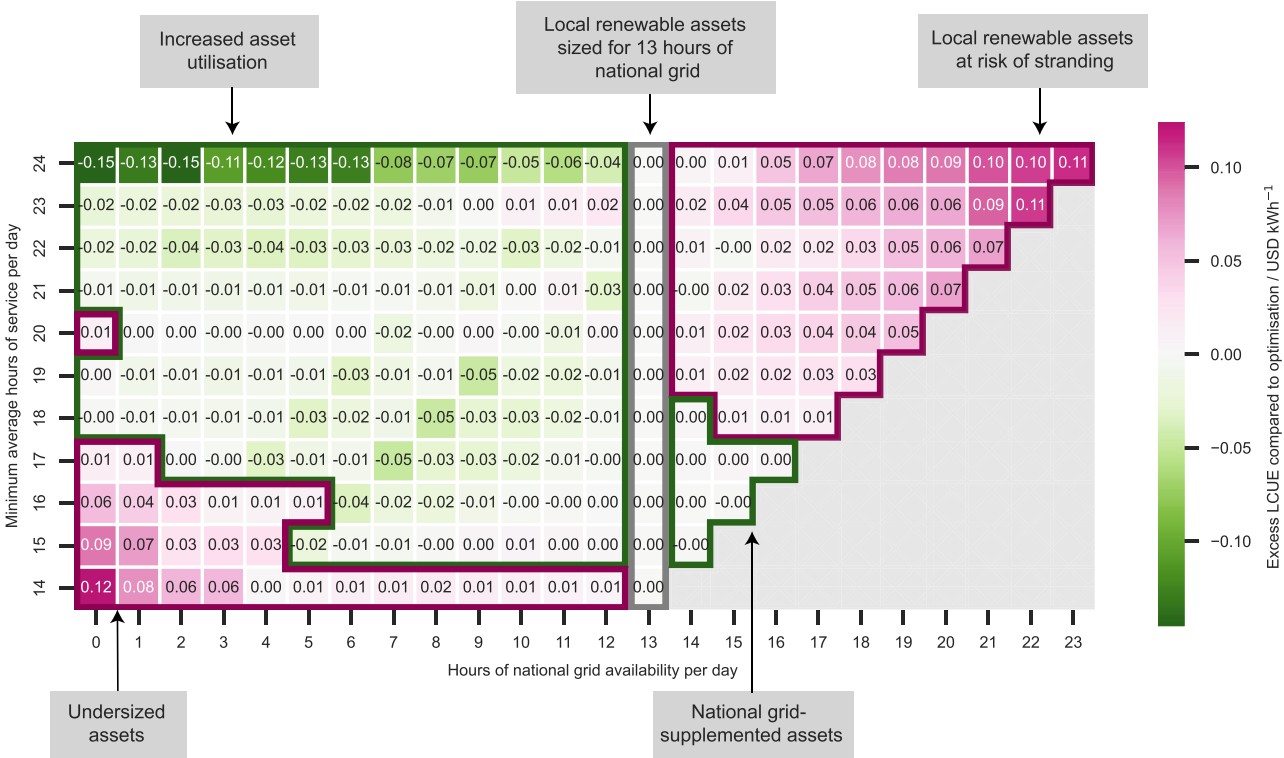

**Fig. 4 | For systems sized to meet a desired level of service given 13 h of national grid availability, the impact on the LCUE (measured in United-States Dollars (USD) kWh⁻¹), compared to an optimised system, is shown as the national grid reliability increases (to the right) or decreases (to the left).** Installations where the system sized assuming 13 h of national grid availability are more cost-effective (have a lower LCUE) than an optimised system are shown in green whilst those which have a higher LCUE are shown in pink. Systems are not investigated when the national grid availability surpasses the initial minimum service requirements due to the difficulty in comparing the costs of national grid-only systems to systems with a small number of installed assets.

whether reductions in the availability of national grid electricity happen during crucial hours such as those with high demand or at inconvenient times which require large storage capacities to satisfy the demand. This effect also applies to those mini-grid systems which are oversized. The "solar-alignment" (how well the hours at which national grid service is lost match with PV electricity generation) of improvements in national grid service impacts the gradient and rate of increase in the LCUE premium.

National grid reliability is an important factor for developers to consider when they are sizing mini-grids that will interconnect with the national grid. The results suggest that, if underestimating the reliability by only 1–2 h, the impact on the cost of energy (Fig. 4) is minimal and therefore there exists a lower risk of wasted investments by the developer. There are two situations where the LCUE would be driven up significantly and investments in mini-grid infrastructure put at risk (highlighted in darker pink areas in Fig. 4). The first is when the national grid's reliability increases substantially (above 18 h) and therefore mini-grid assets are underutilised, representing a reduction in the return of investments from the local assets. The second situation is when the national grid reliability drops to very low availability (less than 2 h), heavily constraining the total energy supplied and therefore driving up the unit cost of energy (see "Methods").

Conversely, there are two situations in which the LCUE would decline or remain stable with changes in the national grid reliability (highlighted in green areas in Fig. 4). In the first situation, there is increased asset utilisation, and a reduction in the availability of the national grid results in higher utilisation of mini-grid assets to meet demand. The second corresponds to the case when mini-grids are supplemented by the national grid; when there is a level of service between 15–18 h, and the national grid improves in reliability (14–16 h), it can complement energy supplied by the mini-grid. For the increased-

utilisation region, whilst the LCUE has decreased slightly due to the reduction in national grid service hours, the overall amount of electricity supplied to the system has also decreased. The service hours provided by the systems shown in Fig. 4 are shown in Supplementary Fig. 4; see Supplementary Note 4.

### Sensitivity analysis of changes in demand on LCUE and the stranded asset risk

Continuing with the perspective of mini-grid developers, we model scenarios that explore the impact of demand loss or gain on the system that may occur alongside improvements in national grid reliability. We do this by sizing the mini-grid infrastructure to provide 16 h of service but including an assumption of present national grid reliability (13 h) and making assumptions about demand change over the 8-year time horizon (see "Methods"). These scenarios are based on the evidence that consumers are only willing to pay a premium for mini-grid electricity if it provides higher reliability[44]. If national grid reliability increases, customers may shift away from paying for higher reliability on the mini-grid infrastructure as the gap between service levels on the mini-grid and national grid falls. This could lead to declines in revenue for operators, placing at risk the prospect of recouping asset investment and leading to stranded assets. If national grid reliability remains stagnant, we also model scenarios where we assume that demand either remains the same or increases (Fig. 5).

Our results show (Fig. 5) that relatively small increases (1–3 cents kWh⁻¹) in LCUE are seen even with a demand loss of 20% or 50% over the modelling time horizon. A reference scenario where the demand and grid availability do not change over the modelling period is shown in grey in Fig. 5 with an average LCUE of 0.21 \$ kWh⁻¹. Although our analysis does not examine the economic viability of the system from the investment side, these results indicate that mini-grid

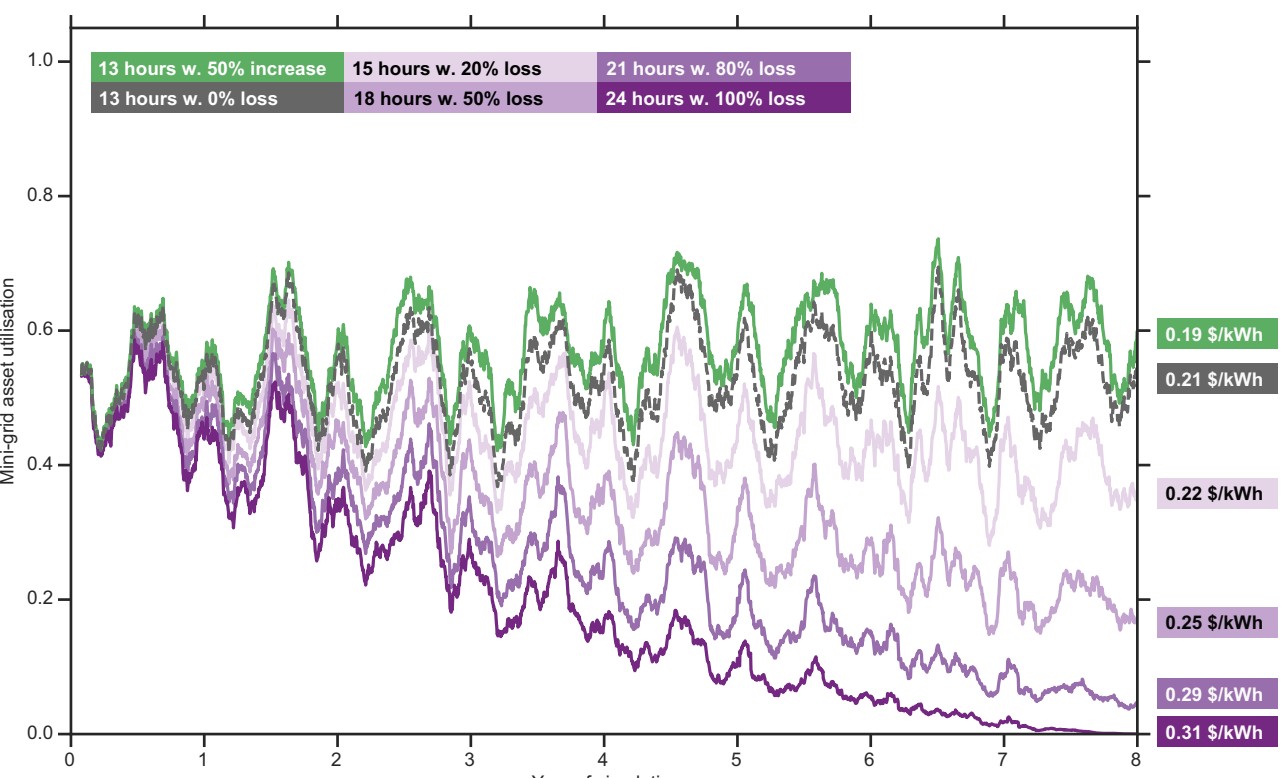

**Fig. 5 | Utilisation of mini-grid assets (the fraction of energy generated by PV installed which was used to meet demand) over time for scenarios where the reliability of the national grid increases or remains stagnant and demand either increases or decreases.** Shown are scenarios for a 50% increase in demand (in green), no change in demand and no increase in national grid reliability (in grey, for reference); 20% electricity demand decrease with an improvement to 15 h of service (in light pink); 50% reduction in electricity demand with an improvement to 18 h of service (in pink); an 80% decrease in electricity demand with an improvement to 21 h of service (in purple), and an electricity demand decrease of 100% with an increase to 24 h of national grid service (in dark purple). The plot shows the mini-grid asset utilisation over the modelling time horizon for each scenario. The coloured blocks on the right-hand side of the figure are the average LCUE for each of the scenarios. Systems shown were originally sized to meet 16 h of service with a national grid availability of 13 h per day (on average), operating under a strategy where electricity provided by the national-grid network is consumed first.

operators may be able to tolerate some demand loss from customers switching to other infrastructure. In more extreme cases, e.g., an 80% reduction in demand with an increase in national grid reliability to 21 h of service per day, the resultant increase in the unit cost of electricity (LCUE) over the 8-year modelling period is 6–7 cents kWh⁻¹. Whilst this is more significant and would be more likely to impact the project viability, given the recent declines in reliability, indicated by official figures[34], it seems unlikely that reliability would increase that rapidly over the project time horizon in rural India. However, project developers need to be aware of the consequences and increased risk of stranded assets. We also consider an increase in demand by 50% on the system in terms of its impact on both the utilisation and the LCUE of the assets installed. Due to the nature of the national grid-prioritisation for sourcing electricity, an increase in demand results in a noticeable but small impact on both the utilisation and LCUE, with the latter decreasing by only 0.02 $ kWh⁻¹. Further work is needed to better understand at what point consumers would be likely to switch, and what the impact on project viability would be in these circumstances.

## Discussion

National grid networks provide chronically unreliable electricity for many rural users in low- and middle-income countries[16], including India[33,34,63] where a significant proportion of the population also remain without a connection[30,64]. Supply can be unpredictable, and, in many cases, is also highly carbon-intensive. Whilst the established strategy for providing access to electricity is to strengthen and expand national grids, this can be expensive on a per-connection basis in lower-density

rural areas that are further from urban centres[16]. In response to these challenges, we set out to address the following research questions: first, to what extent can national grid–connected solar PV mini-grids improve electricity service hours for consumers across different reliability scenarios, and what are the costs and emissions of this approach? Second, how do future potential changes in national-grid reliability and consumer demand affect the risk of asset stranding and the long-term financial viability and environmental sustainability of solar PV mini-grids? And, third, what policies can be implemented in order to better mitigate against the risk of asset stranding and better facilitate the interconnection of mini-grids with the national grid?

We find that local renewable mini-grid networks may be able to boost hours of supply for rural users whilst requiring lower investments than national-grid extension. We find that doing so can also simultaneously reduce emissions across a range of grid reliabilities and levels of service provision. We find that there is a risk of asset stranding when the reliability of the national-grid network improves. When quantified, however, the increase in the LCUE is minimal compared to the tariffs commonly charged by mini-grid developers.

The private sector plays an active role in mini-grid development in many countries with lower rates of electricity access, often filling a gap left by the state that has failed to improve reliability and connectivity via grid extension. To realise the potential benefits described above, whereby mini-grids can work in tandem with unreliable grids, a range of policy interventions are needed to encourage and de-risk investment in rural areas that may be presently or soon connected and served by national grid networks. Firstly, technical legislation is

needed, such as the agreements of connection and distribution standards to facilitate mini-grids connecting to national grids and rules that permit mini-grids to operate under islanding conditions when the national grid is unavailable[20–24]. Countries such as Nigeria, Cambodia, and Kenya have established clear technical grid-compatible standards for the mini-grid. This compatibility allows mini-grids to integrate smoothly with the main grid and enables distribution companies to purchase the assets if expansion occurs during the mini-grid's operational lifespan.

Secondly, clear policies are needed to give revenue certainty to developers and investors, such as guaranteeing the supply of customers at specific tariff levels to ensure a return on their investment. This second set of policies is fundamental when considering how national grid reliability might improve over time and potentially whether consumers may switch away from mini-grid infrastructure as a result. Our results demonstrate that a moderate demand loss does not seem to drastically impact the unit cost of supply: a 15% increase in the LCUE is seen when there is a 50% reduction in demand over the 8-year time horizon (when the grid improves by 50% to 18 h of availability). Investments are more likely to be at risk due to customers switching to only the national grid, particularly at higher levels of demand loss resulting from an improved grid. On the one hand, our results highlight that investments may be insulated from demand loss resulting from some loss of customers, with mini-grid providers potentially able to absorb these costs. It is vital to provide investors with protection against potential demand and revenue losses in the case of significant improvements in national grid reliability and associated consumer switching.

Policies to protect mini-grid revenues could exist in a variety of forms. For example, the exclusive right to supply a given set of customers within a zone designated for grid-connected mini-grids for a minimum length of time. In countries like Nigeria and Tanzania, policies such as exclusive permits for developers and commercial agreements for generation and distribution assets—like tripartite agreements among distribution companies, communities, and operators—have enabled private investment in this sector to provide reliable access to electricity in rural communities[6]. The competitiveness of mini-grids can be improved by providing incentives like subsidies to developers for reliable electricity supply or penalising distributors for unreliable supply of electricity[14]. Alternatively, and beyond the scope of this study, facilitating mini-grids to also sell energy back to the grid could further improve operating revenues. Providing more explicit guidance on the technical and regulatory requirements for interconnection at the state and national levels could support private or non-governmental actors to develop the equipment, operating strategies and business models required to operate local systems. Legislation that strengthens years of exclusive supply can allow developers to capitalise on improved national grid reliability whilst being shielded from the risks of customers switching over to a national grid-only connection. This could both increase the availability of power and quality of service to rural communities and support national electricity access plans in low- and middle-income countries.

A key consideration beyond the regulatory environment is that of the technical components (and costs) involved in mini-grid to national grid connection. National-grid connectivity adds complexity to mini-grid systems, as well as additional costs. Our study prioritises the use of lower-cost electricity from the national grid (see "Methods"); however, in practice, control systems may be needed to determine under what conditions the mini-grid should prioritise the use of the national grid, local solar generation or stored electricity, depending on the power purchase agreement. Whilst we consider the costs of components needed to import electricity from the grid (see Supplementary Table 3), a more in-depth consideration of related power system aspects is beyond the scope of this study. Future research should consider this in more detail, as well as mini-grid export capabilities for

both strengthening national grid networks and improving the financial viability of mini-grids.

Considering the specifics of the case study location, existing grid-based electrification strategies do not yet provide a reliable supply of electricity in rural areas of Uttar Pradesh[33,34]. Stakeholders should consider implementing systems that are designed to provide modest service improvements, which are also resilient to improvements in grid availability. Providing an additional 4 h above the average grid availability of 13 h in Uttar Pradesh provides both an increase in availability and a decrease in LCUE (Fig. 3d). These systems could be affordable for rural communities, as our results show that the overall tariffs are comparable to or lower than those currently charged by mini-grid developers in India, and the systems that are designed to offer higher levels of service can result in a higher willingness to pay[43–46]. The current legislation and technical regulations in India are ambiguous for the proposed interconnections[22,23]. Without clear policies, private investment in this area is likely to be limited which in turn will limit the socioeconomic development of these rural areas.

Finally, national grid-connected mini-grids offer a route to not only improve the reliability of but also decrease the carbon intensity of electricity supplied. Our results demonstrate that, apart from a very high national grid reliability situation, solar mini-grids interconnecting with the grid offer substantial emissions intensity reductions when considering our example of the Indian grid which is aligned with India's national climate action plans. Whilst consumers at the bottom of the energy ladder contribute only a tiny proportion of annual global GHG emissions, growth in demand over time could change this if low-income countries don't leapfrog to low carbon energy infrastructure[16]. Moreover, given the need for low- and middle-income nations to both meet climate obligations and rapidly improve the number and quality of electricity connections, renewables-based mini-grids interconnecting with national grids could offer a partial solution to these dual challenges.

Our study is based on an example from India, a country with challenges relating to the reliability of its national grid network and connectivity of all households. However, the analysis and findings we present are relevant for other countries that have national grids with low reliability and high carbon intensity. This applies to many countries in Sub-Saharan Africa, which aim to increase access to electricity via both national grid extension and distributed renewable generation in line with their sustainability targets. Emerging opportunities for climate financing, renewable energy credits, and carbon prices would likely further support the deployment of distributed renewables in India and elsewhere and could be the focus of future investigations.

## Methods

### System characterisation and modelling

We use the open-source energy-system modelling framework "Continuous Lifetime Optimisation of Variable Energy Resources" (CLOVER)[56,57] to simulate the performance of systems capable of providing electricity to a rural community in Uttar Pradesh, India. We use CLOVER to investigate systems consisting of a combination of solar PV collectors, battery storage and a national grid connection. Similar to other energy systems models[65], the CLOVER modelling framework uses electricity demand and supply data to simulate energy systems at an hourly resolution, and, for a given optimisation criterion, determines the optimum system. The CLOVER software—the operation of which is discussed in more detail throughout the methods—is open-source, customisable and many combinations of potential implementation scenarios can be modelled with ease, as was done in this study.

In this study, we utilise the open-source energy-system modelling framework CLOVER[56,57] over commercially-available alternatives, such as "Hybrid Optimization of Multiple Energy Resources" (HOMER) (https://www.homerenergy.com/). This choice enables us to adapt the

software to model diminishing demands and improvements in grid reliability over time—to investigate the notion of asset stranding—as well as enabling transparency and reproducibility of the results produced so that future researchers can fully scrutinise the methods employed in the study.

We characterise the system by its generation (its installed PV and battery storage capacities and the availability of the national grid) and its demand (the load profile over a given time period). We simulate systems over 8 years: the target time frame for achieving the United Nations' SDG 7 and other national sustainability targets[51]. Technical parameters of the system components considered—including the batteries, PV collectors, and distribution network—are given in Supplementary Table 2.

## Load profiles
Similar to other studies[58,66–69], we generate stochastic load profiles for each type of appliance in the community. We categorise appliances as either domestic or commercial and further break this down by device type (Supplementary Table 1). We determine the number of appliances of each type in use at any given time by randomly selecting a value from a binomial distribution based on the total number of appliances in the community and the probability that any given appliance is in use.

For the appliance ownership values and usage probabilities, we take values from the energy surveying work presented by Agrawal et al.[32]. Their report gives representative values for device ownership and utilisation for a community of the size of a typical village in rural India. The community size is scaled from the national-average value (860 households) used in Agrawal et al.[32] to the average community size they presented for Uttar Pradesh (547). We take the average appliance ownership and power as given. We validate the values obtained against other published literature[43,58,69].

Using a community size of 547 households, 66 enterprises, and one flour mill, our case-study community has a total demand of 761 kWh day$^{-1}$. We do not consider the use of electricity for agricultural water pumping as the land requiring irrigation is likely to be dispersed and far away from the community, making standalone solar or diesel pumps more relevant. Future work may consider these productive uses as a means to lessen the risk of asset stranding.

## Solar generation
We source the solar energy-generation data from Renewables.ninja[70]: a web interface which uses the Global Solar Energy Estimator[71], weather data reanalysis models[72,73], and satellite observations[74] to estimate the electricity output of one unit of PV (1 kW$_p$) installed at any point on the Earth's surface. We take data for our chosen location, Bahraich District in Uttar Pradesh, over the period from the 1$^{st}$ of January 2007 to the 31st of December 2017. We assume that there is sufficient area for the PV capacity to be installed in a location absent of any local shading, i.e., that the data obtained from Renewables.ninja[70] does not need any adjusting for local climatic factors. Information about the PV system we model is given in Supplementary Table 2.

## Grid availability
Data for the availability of the grid is taken from the Energy Supply Monitoring Initiative from Prayas (Energy Group)[62] which uses remote monitoring devices to record the availability and quality of grid electricity at hundreds of locations around India. We use data from a representative location with adequate data availability[63], Bhawaniyapur, in a rural area of Bahraich District. The location received an average availability of 12.5 h of grid electricity per day, similar to the 12-h average for the state at the time the data were collected[32].

We transform the proportion of minutes of electricity available in each hour from this monitored profile into an hourly profile for the probability of the grid being available (and able to meet 100% of demand) at any given hour over 1 year, $G_P^{12.5}(t)$. We then mask this

probability profile to yield new profiles by adding and removing hours of grid availability, $G_P^h(t)$. Doing so incrementally yields 25 profiles from 0 to 24 h of availability on average over 1 year. This allows us to maintain the relative hourly, daily, and seasonal fluctuations in the monitored data whilst varying the overall reliability. For most of the analysis, we assume that the availability of the grid network remains constant throughout the modelling period: using a fixed availability profile, $G^h(t)$, for each scenario. When investigating stranded assets, we consider increases in the availability over time, which is discussed in more detail below.

## Energy balance and usage strategies
The model assesses the energy balance in the system at hourly intervals by comparing the load placed on the system to the available energy sources. The energy-balance strategy either uses electricity from local renewable assets first with the national grid connection as a backup ("local generation prioritisation") or energy from the grid first with local assets supplementing the connection ("grid prioritisation"). The approach is shown in Supplementary Fig. 2 (see Supplementary Note 2).

Under both scenarios, if load energy remains after all supply options have been exhausted then a blackout is recorded. If the load is met and excess PV energy remains then the surplus is used to charge the batteries if not already fully charged. If there is still excess PV energy that has not been utilised, this is recorded as "dumped" energy.

## Technical performance assessment
The open-source modelling framework, CLOVER[56,57], provides metrics for assessing the performance of a system. We use the notion of "blackouts", where a blackout is an hour during which any amount of electricity demand went unmet (i.e., any shortfall of energy results in the hour being classified as a blackout). We define the availability of the system as the percentage of hours for which blackouts did not occur and use this as the threshold condition for the suitability of a system. Multiplying by 24 h yields the "minimum service hours" per day, $\bar{h}_{min}$:

$$\bar{h}_{min} = 24 \times \left(1 - \frac{\sum_t^{N_h} B(t)}{N_h}\right), \qquad (1)$$

where $B(t)$ is whether a blackout occurred in an hour (a value of 1) or not (a value of 0) and $N_h$ is the number of hours that were simulated. An advantage of using blackouts as a metric of reliability is that this most closely aligns with evaluations of mini-grid and national-grid performance undertaken by surveyors and governments[32]. However, it fails to account for the demand in each hour, potentially reaching high levels of minimum service whilst meeting only a small fraction of the demand placed on the system.

## System assessments
We perform two types of assessment on the system: financial and environmental. The financial assessment concerns the costs associated with the system, whilst for the environmental assessment we limit ourselves to the GHG emissions—both embedded emissions and those arising from the consumption of national grid-sourced electricity. We do not consider the full life cycle GHG emissions of the national grid electricity. As a result, our values of national grid emissions impact are underestimated. Our analysis does not account for the use of kerosene as an alternative to electric lighting, despite the CLOVER model having the functionality to include it. This is due to complexities and uncertainties in estimating its use and impacts. This omission represents a limitation of our study. If kerosene were to be included, it would be likely to further improve the emissions savings of switching to the mini-grid over all hours of grid availability and service provision[52,58].

---

**Algorithm:** Determine the optimum system $\psi(B,P)$ given a minimum service level $h_{\min}^{target}$

---

**Require:** $h_{\min}(\psi) \geq h_{\min}^{target}$
**Inputs:** $B_{\max}$, the initial maximum battery capacity permissible,
  $B_{\min}$, the minimum battery capacity permissible,
  $h_{\min}^{target}$, the minimum permissible average number of service hours,
  $P_{\max}$, the initial maximum PV capacity permissible,
  $P_{\min}$, the minimum PV capacity permissible.

1:  **while** $h_{\min}(\psi) < h_{\min}^{target}$. **do**
2:      $B_{\max} \Leftarrow B_{\max} + \Delta B$
3:      $P_{\max} \Leftarrow P_{\max} + \Delta P$
4:      Simulate $\psi(B,P)$
5:  **end while**
6:  $B_{\max} \Leftarrow B_{\max} + \Delta B$
7:  $P_{\max} \Leftarrow P_{\max} + \Delta P$
8:  **for** $B := B_{\max}$ **to** $B_{\min}$ **step** $-\Delta B$ **do**
9:      **for** $P := P_{\max}$ **to** $P_{\min}$ **step** $-\Delta P$ **do**
10:          Simulate $\psi(B,P)$
11:          **if** $h_{\min}(\psi) < h_{\min}^{target}$ **then**
12:              Go to line 9
13:          **end if**
14:      **end for**
15:  **end for**
16:  **for** $\psi(B,P)$ **do**
17:      Calculate LCUE$(B,P)$
18:  **end for**
19:  $\{\psi_{best}\}(B,P) \Leftarrow \arg\min_{\psi(B,P)}[\text{LCUE}(B,P),\{B,P\}]$
20:  $\psi_{best}(B,P) \Leftarrow \arg\min_{\psi(B,P)}[B\{B\}]$
21:  $\psi_{best}(B,P) \Leftarrow \arg\min_{\psi(B,P)}[P\{P\}]$
22:  **if** $B = B_{\max}$ **or** $P = P_{\max}$ **then**
23:      $B_{\max} \Leftarrow B + \Delta B$
24:      $P_{\max} \Leftarrow P + \Delta P$
25:      **for** $B := B_{\max}$ **to** $B_{\min}$ **step** $\Delta B$ **do**
26:          Simulate $\psi(B,P_{\max})$
27:      **end for**
28:      **for** $P := P_{\max}$ **to** $P_{\min}$ **step** $\Delta P$ **do**
29:          Simulate $\psi(B_{\max},P)$
30:      **end for**
31:      Go to line 16
32:  **end if**
**Output:** $\psi_{best}(B,P)$ such that LCUE$(B,P)$ is minimal.

---

**Fig. 6 | Algorithm for determining the optimum system, $\psi(B,P)$ given a minimum service level $h_{\min}^{target}$.** The algorithm works by iteratively searching through to determine systems which meet the required service level and then exploring the parameter space in terms of installed battery, $B$, and photovoltaic (PV), $P$, capacities.

**Financial assessment.** We used the "levelised cost of *used* electricity" (LCUE) to assess the economic performance of the system. This differs slightly from the more commonly-used "levelised cost of electricity" (LCOE) by explicitly only accounting for the electricity used by the community, discounting electricity that is generated but not used and which therefore is of no benefit to the community. Many studies and models available use LCOE as all generated electricity is used, but we make this distinction clear to explicitly account for overgeneration.

The LCUE of a system is given by its discounted total costs, $C_{tot}$, divided by the total discounted energy it supplied to the users, $E_{tot}^{disc.}$, such that

$$\text{LCUE} = \frac{C_{tot}}{E_{tot}^{disc.}}, \qquad (2)$$

where

$$C_{tot} = \sum_{n}^{N_y}\left(\sum_{j}\frac{C_n(j)}{(1+r)^n}\right), \qquad (3)$$

where $C_n(j)$ denotes the cost of component $j$—which could be equipment (capital expenditure, CAPEX) or maintenance (operational expenditure, OPEX)—occurring in year $n$, $N_y$ is the number of years that were simulated, and $r$ denotes the discount rate. The discounted

energy is given by:

$$E_{tot}^{disc.} = \sum_{n}^{N_y}\frac{E_n}{(1+r)^n}, \qquad (4)$$

where $E_n$ denotes the undiscounted energy supplied in year $n$.

**Environmental assessment.** We quantify the environmental impact of a system as the total GHG emissions over its lifetime. We calculate this by summing over the emissions from each component for each year of the system's operation:

$$\text{GHG}_{tot} = \sum_{n}^{N_y}\left(\sum_{j}\text{GHG}_n(j)\right), \qquad (5)$$

where $\text{GHG}_n(j)$ denotes the emissions attributed to component $j$ in year $n$. We normalise these cumulative emissions by considering the emissions intensity:

$$\text{Emissions intensity} = \frac{\text{GHG}_{tot}}{E_{tot}^{Undisc.}}, \qquad (6)$$

to give a value per unit energy consumed. Note that we use the undiscounted electricity, $E_{tot}^{Undisc.}$, by convention. The emissions intensity of grid electricity linearly decreases from a present value of

**Table 1 | Scenarios explored when investigating increases in the reliability of the national-grid network alongside changes in demand**

| Scenario | Percentage change in demand/% | Final grid reliability/h day⁻¹ |
|---|---|---|
| Increasing demand | 50 | 13 |
| Static demand | 0 | 13 |
| Decreasing demand | 20 | 15 |
|  | 50 | 18 |
|  | 80 | 21 |
|  | 100 | 24 |

632 $gCO_2$eq $kWh^{-1}$ to 440 $gCO_2$eq $kWh^{-1}$ at the end of the 8-year modelling period[3,41,61]. Although the electricity generation mix in Uttar Pradesh is particularly coal-heavy[75], we use the Indian national grid emissions intensity values in preference to values for Uttar Pradesh given that transmission of electricity occurs between different parts of India's grid[76].

## Optimisation process
We define the optimum system to be that which provides the desired level of availability ($h_{min}$) at the lowest LCUE. We use an iterative heuristic search algorithm (Fig. 6) to test the performance of each system. A prospective system, $\psi(B, P)$ is simulated over the lifetime of the system and its impacts are calculated. The algorithm searches, based on some initial guess at the capacities ($B_{max}$ and $P_{max}$ for the battery and PV capacities), for a system which provides the required level of service.

The algorithm then simulates all systems in decreasing capacities, looping through until systems no longer meet the required minimum level of service. The LCUE for these systems is calculated and the system with the lowest LCUE is selected. If the optimum system had either the largest battery or PV capacity of the systems simulated, a search is undertaken to see whether increasing the system capacity could reduce the LCUE of the system.

Similarly to other studies[77], the LCUE is a convex function along "isoreliability curves"–curves with the same level of reliability–and so we take the optimum system to be that which provides the required availability at the lowest LCUE with the smallest capacity of installed components as this is most akin to the perspective of a system developer. We start the search with capacities of zero and use step sizes of $\Delta P = 5$ $kW_p$ and $\Delta B = 5$ kWh.

In our analysis, we investigate each combination of minimum service hours and grid hours: $h_{min}, G^h(t) \in \{0, 1, 2, ..., 22, 23, 24\}$. This range encompasses scenarios with levels of service and grid availability below those currently experienced on average but which are commonplace for many rural communities[32] as well as those representing an improved level of service in the future.

## Assessing stranded assets
We assess the potential for assets to become at risk of being stranded by comparing the techno-economic performance of mini-grid systems sized for the current level of grid reliability but operating under a different grid profile. Keeping the solar PV and battery capacities the same, we consider systems that are optimum under $G^{13}(t)$ and simulate them under all grid-availability profiles, $G^h(t) \forall h$. This allows us to assess the performance of systems designed for the current grid reliability level from the point of view of both mini-grid developers and installers, as well as policymakers, in the cases that the grid availability either improves or is worse than expected.

Initially in our study, we consider the performance of systems sized for 13 h of grid availability operating over 8 years under different availabilities of the national grid from start to finish. Whilst this does not include the impact of gradually increasing reliability, it provides an

overview of the performance of these systems as the reliability of the national grid changes, including the additional average annual service hours provided and the impact that this, and the improved national grid-reliability, have on the LCUE. We also consider scenarios where the reliability of grid electricity increases linearly over time. For these scenarios, we assume that an increase in the reliability of the national grid network would prompt customers to switch over from the mini-grid to using cheaper power sourced directly from the national grid. We consider representative scenarios for a linear decrease in demand alongside a linear increase in the reliability of the grid, as well as a linear increase in demand with constant national grid reliability. These are detailed in Table 1. Potential second-order effects, whereby the increase in the LCUE resulting from a decrease in asset utilisation results in further demand loss, is not considered here. Additional scenarios with varying degrees of demand loss for fixed levels of grid reliability are outlined in the Supplementary Information.

## Data availability
The output data from the CLOVER optimisations and the code used to process it for this study are available on Zenodo at https://doi.org/10.5281/zenodo.15808177.

## Code availability
The CLOVER model code used to run the study is available at https://github.com/CLOVER-energy/CLOVER and has documentation available at https://github.com/CLOVER-energy/CLOVER/wiki with worked examples of using the model to analyse a case study system.

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

## Acknowledgements

This work was supported by UK Research and Innovation (UKRI) [grant number UKRI314, J.N.], the UK Engineering and Physical Sciences Research Council (EPSRC) [grant numbers EP/T028513/1 (ATIP), J.N., EP/R045518/1, J.N. and EP/X52556X/1, B.W., H.B. and J.N.], the UKRI Frontier Research Gurantee [grant number EP/Z533361/1, J.N.] and the UK Natural Environment Research Council (NERC) [grant numbers NE/S007415/1, B.W. and H.B.]. This work was also supported by the San Francisco not-for-profit ClimateWorks Foundation. The work was also supported by the Royal Society under an International Collaboration Award 2020 [grant number ICA\R1\201302, C.N.M.]. H.B. and B.W. would like to gratefully acknowledge the support of NERC and the Grantham Institute—Climate Change and the Environment for Ph.D. scholarships. S.M. would like to acknowledge support from the Research Council of Norway project Accelerating Climate Action and the State: Getting to Net Zero (ACCELZ), project number 335073 and the European Commission Horizon Europe programme "IAM COMPACT" under Grant Agreement No. 101056306. J.N. thanks the Royal Society for the award of a Research Professorship and the European Research Council for award of an Advanced Grant (grant no. 742708, CAPaCITy).

## Author contributions

P.S. and J.N. conceived the original study; with updates to the study design and analysis conceived by H.B., B.W. and S.M. Code for completing the study was written by P.S. and B.W. The analysis was performed by P.S. and B.W. Analysis of the results was done by B.W., H.B. and S.M. The visualisation was done by B.W., H.B. and P.S. The writing and editing of the manuscript was done by P.S., H.B., B.W., S.M., J.N. and C.N.M.

## Competing interests

The authors declare no competing interests.
