## [Transparent Peer Review file · Nature Communications]

Opportunities for decentralised solar power to improve reliability, reduce emissions and avoid stranded assets

Corresponding Author: Professor Jenny Nelson

Version 0:

Reviewer comments:

Reviewer #1

(Remarks to the Author)

The tension between centralized and decentralized modes of organizing the electricity generation capacity has been debated for several years. Its a proposition valid for both households who already have access and for those who don't. Decentralized generation has been explored to provision access through pilot installations, with much of the up-front cost being subsidized through grant aid. In absence of the grant aid the unit cost of generation are high. Such installations tend to suffer from lack of maintenance and reliability is often low. How or if such a model can be scaled has been a topic of much discussion. There are installations which recover their costs (plus profit) by charging tariffs which are higher than the grid as has been indicated in the surveyed literature.

Usually, the last-mile connectivity is provided by the state utility, and the paying capacity of the Residents in these areas is low.

Decentralized distributed generation for areas with access is also important component of the strategy to maximize deployment of renewable energy. This strategy has its own challenges, including grid management, utility death spiral etc. In the framing of the problem, mini grids contribution to enhancing the affordability is included. This follows arguments and evidence on electricity usage enhancing the enterprise activity and hence the paying capacity, while also underscoring that mini-grids prices are higher than the grid, nonetheless people pay for it in absence of alternatives, indicating the value the customers place on reliable electricity. The claim in the following sections is of mini grids enhancing the affordability, and its not clear if its because of enhanced enterprise activity. In the modelling it becomes clear that this effect is not included and still the affordability increases across range of scenarios, which is counter-intuitive.

In subsequent explanations, it comes out that this is on account of the additional cost of expanding the grid connections.

Page 6, line 101-7: For national grid the additional cost of distribution network for the grid is socialized over larger network. Recovering this cost from the electricity sold to the local community is not the practice. This seems to be main driver of the results, though it is not discussed sufficiently and justified.

Separately, have authors come across reliability reports for mini grids? I have come across some reports of neglect and lack of upkeep for a few such mini grid installations.

Figure 2b) The legend should be labelled battery capacity.

In results section:

“Regardless of the national grid availability level (in hours of service provided), increasing the minimum service hours by similar amounts (e.g., by two, from four to six hours per day) requires similar increases in PV capacity.”

The increase in PV capacity qualified by “similar”. Could the authors clarify the similarity? PV capacity increases for increasing service hours, the quantum of increase is different. The increase is not proportional either.

Page 6, line 101-7: 110-12: Grid emission intensity will change with increasing centralized solar and wind installations.

Reviewer #2

(Remarks to the Author)

The authors of the manuscript provide interesting and important research related to rural electrification strategies in developing countries. This research is relevant and timely with the governments around the world tackling issues of climate change and providing electricity to unelectrified rural villages. This research addresses/compares mini-grids, grid-based electricity and a combination of both to provide rural electrification strategies to avoid stranding of mini-grid assets and

reduce emissions.

Here are some general comments:

- The authors could do a brief literature review on the previous related research on this topic so that the readers get a context of the value add provided by this research.

- It is mentioned in the title as well as across the manuscript that one of the aims is to reduce emissions. The authors consider a linear decrease in emission intensity of grid-based electricity from a present value of 632 gCO₂eq/kWh to 440 gCO₂eq/kWh at the end of the eight-year modelling period. Supplementary Table 3 gives the numbers for the mini-grid. The sum of all GHGs – 1158 gCO₂eq/kWh. A few comments and questions on this:

- o Why is the unit of 'kWh' not used for mini-grid systems for the emissions intensity calculation?

- o Do the authors consider the complete life cycle emissions for the grid electricity as well?

- o Why the consideration of linear decrease in emission intensity of grid-based electricity? What are the assumptions for this linear decrease?

- o Uttar Pradesh has close to 90% share of coal in its electricity generation. I assume the emission intensity will be higher than 632 gCO₂eq/kWh. Please check the following reference <https://cea.nic.in/cdm-co2-baseline-database/?lang=en>

- Financial and technical assumptions: Do the authors consider changes in technical and financial parameters across the 8-year period? The assumptions in Supplementary Tables 2 and 3 are for which year? Please specify.

- Do the authors consider the influence of increasing standard of living till 2030 on appliance ownership and in general on the load profile? Please explain.

- Could the authors explain more why LCUE was used instead of LCOE? Shouldn't the cost of overgeneration (curtailment) be part of the cost calculation? If LCOE was used instead of LCUE, how would it affect the results?

Specific comments

Abstract

- Please clarify if you are dealing with 'electricity' or 'energy'. Line 1 gives reference to 'electricity', while line 2 refers to 'energy' infrastructure.

- 'We explore the impact of reduced demand to quantify...' Is it 'energy' or 'electricity' demand. Please clarify to the readers.

Introduction

- Line 20: 'National grid electricity tariffs...' This statement is not true for all countries globally, consider the case of Australia as an example. There could be other countries as well. Check this publication <https://www.sciencedirect.com/science/article/pii/S1364032121001581>. Your references for the sentence are for India. So, please mention that the sentence is valid for a group of specific countries or India.

Methods

- Line 377: The authors assume that '...an increase in the reliability of the national grid network would prompt customers to switch over from the mini-grid to using cheaper power sourced directly from the national grid'.

- o What is the basis for such a strong assumption? The authors are assuming that grid-based electricity will still be cheaper than mini-grids even in 2030. Which might or may not be true. With the dramatic cost decline observed in the last decade for solar PV and Li-ion batteries, this might not hold true.

Results

- Line 87-88: Did you consider the revenue from selling excess electricity in the modelling? If so, what is the compensation for the sold electricity in \$/kWh?

- Figure 3a,3b,3c: Please have the units on the left y-axis in brackets.

- Figure 2: The authors haven't mentioned about the assumption for kerosene lamps. Are they being used in any hour? Are the costs of purchasing fuel (kerosene) considered?

- Line 114-115: From Figure 2c, relying solely on the grid provides the cheapest LCUE – 0.16 \$/kWh. Please explain if the interpretation is wrong.

Reviewer #3

(Remarks to the Author)

1. Key results

- Improvement in the reliability of the national grid and subsequent longer hours of reliable electricity supply from the grid increases the levelized cost of used electricity. The tariff charged by mini-grid companies can cover the increase if the improvement is less than 50% and there are no critical times when both the national grid and the mini-grids cannot meet the demand.

2. Validity

- The scope of stranded assets is less clear. Specify if it indicates only mini-grids (transmission and distribution systems) or includes both grids and decentralised solar generation. Given that the audience familiar with this research field knows the

stranding of fossil fuel assets and the death spiral well, the precise definition is indispensable for accurate interpretation.

- The scenario of a 50% increase in demand with the same reliability looks like speculation and thus needs rationale or supporting evidence. The reviewer sees associated increases in demand with improvements in the national grid reliability make sense in the context of rural areas. There is plenty of evidence that an unreliable electricity supply has restricted purchases of electronic appliances such as fridges, cleaners, washing machines, and microwaves.

3. Significance

- The findings could provide a practical transition strategy for electricity systems in countries where the government prioritises rural electrification while incumbent grids are featured by coal lock-in.

4. Data and methodology

- Table 1 does not support the linearity between hours of operation and percentage changes in demand. The audience cannot understand why a 2- or 3-hour longer access to the national grid operation leads to incremental demand loss by 20% or 30%.

- Figure 4 shows 1-3 cents increase in LCUE with a demand loss of 20% or 50% instead of 1-2 cents in the main text (LL.182-3).

5. Analytical approach

- The different referential points in the two scenario analyses devalue the results. The first analysis sets the reference point at 8 hours of grid availability (Figure 2&3), and the second at 13 hours (Figure 4&5). The inconsistency makes the results of the first analysis powerless and blocks comparisons of the impacts on carbon intensity between the two analyses.

6. Suggested improvements

- Use the same referential point in grid availability hours for the two scenario analyses.

- Add a sensitivity analysis of the LCUE with mini-grids with and without imposing the cost of attaining India's NDC goal (531 gCO₂eq/kWh on average) for electricity imported from the national grid in the second analysis. Given that mini-grids coupled with decentralised solar generation are installed to increase electricity access without increasing carbon emissions, the comparison should be made on a level playing field. The manuscript would make a far more significant scholarly contribution to the journal Nature Communication.

This improvement does not require new scenario analyses from scratch. It is sufficient to refer to the recent literature on the cost of carbon emissions, estimate it for the electricity imported from the national grid in India, and use the estimated cost for sensitivity analyses.

- The proposal of the exclusive right to supply a given set of customers within a zone designated for grid-connected mini-grids for a minimum length of time makes sense. Authors may refer to institutions of privately invested mini-grids in some countries and provide them as supporting evidence. While it becomes business as usual to conclude long-term power purchase agreements to attract private investment in renewable generation with storage and grid connection projects, the authors counter-argue this proposal in the paragraph starting from 241.

- Authors may discuss transmission and distribution development plans as a driver of improving the national grid reliability. An increasing number of emerging markets and developing economies have implemented plans recently to avoid unplanned installations of generation capacity and subsequent increases in grid development costs. For the case of the Philippines, see Delina et al., Energy Strategy Review 53, 2024, 101407.

Version 1:

Reviewer comments:

Reviewer #2

(Remarks to the Author)

Thank you for revising the manuscript. The manuscript has been revised with appropriate responses and text additions in the article.

I do not have any further comments.

Reviewer #3

(Remarks to the Author)

Dear authors,

The reviewer noted the authors' sincere responses to the feedback and revisions to the manuscript in the main text and supplementary materials. They satisfy the reviewer, except for the structure, organization, and expressions.

1. What are the noteworthy results?

****Yes.**

2. Will the work be of significance to the field and related fields? How does it compare to the established literature? If the work is not original, please provide relevant references.

**** The manuscript does not present past research and research gaps, which makes it difficult for the reviewer to evaluate originality and novelty.**

3. Does the work support the conclusions and claims, or is additional evidence needed?

**** Provide assumptions on the villagers' electricity access when inaccessible to national grid electricity. Do they use charcoal or other biomass-based fuel as alternatives, or live without any electricity services? Do they reduce consumption of these alternatives but keep using them unless they can access electricity for 24 hours with the national grid and local mini-grid combined?**

The assumption may affect the LCUE and emission intensity.

4. Are there any flaws in the data analysis, interpretation and conclusions? Do these prohibit publication or require revision?

**** Interpretation and expressions can be updated. This holds particularly in the section on results and discussion.**

(1) Figures 2c and 2d and their explanation should be placed after the next section elaborating on LCUE. They are too complicated and demanding for the audience to understand the results only with these figures and explanations.

(2) Rephrase the sentence in Lines 115-116 "with only mini-grid systems providing 20 or more hours of service with a national grid availability of 18 or more hours per day exceeding this target" in plain English.

(3) The paragraph starting from Line 204 may be replaced with research questions. It looks like a repetition of the motivations in the Introduction. Reminding research questions is much more productive.

(4) The paragraph starting from line 217 is redundant. It must be consolidated with the introduction and the policy implications starting from line 237. It is odd to see similar policy implications separately.

(5) Lines 229—231 do not make sense from an economic viewpoint. The interpretation may be improved if rephrasing:

1) demand loss from an improved grid substantially reduces revenue for mini-grids, generating stranded costs.

2) the demand loss increases the LCUE only slightly, thus does not generate further demand loss and increases in stranded costs.

(6) Policy implication to India in Lines 264—266 is insufficient. The results reveal that perceived stranded risks make private investors hesitant to invest in local mini-grids. Discuss and suggest policies that can reduce the risks and their perception.

5. Is the methodology sound? Does the work meet the expected standards in your field?

****Methodology is sound. However, the section title of "Scenarios including electricity demand loss and impacts on stranded asset risk" in Line 179 may be replaced with "sensitivity analysis of demand and LCUE" to represent the subsection.**

6. Is there enough detail provided in the methods for the work to be reproduced?

****Yes.**

Response to Reviewers: NCOMMS-24-28162-T

Reviewer 2

#	Comment	Response
1	The authors of the manuscript provide interesting and important research related to rural electrification strategies in developing countries. This research is relevant and timely with the governments around the world tackling issues of climate change and providing electricity to unelectrified rural villages. This research addresses/comparates mini-grids, grid-based electricity and a combination of both to provide rural electrification strategies to avoid stranding of mini-grid assets and reduce emissions.	We thank the reviewer for their comments and are glad they see the research as timely and relevant. We appreciate the time taken to review the manuscript and hope our addressing their comments strengthens it.
2	The authors could do a brief literature review on the previous related research on this topic so that the readers get a context of the value added provided by this research.	We have added additional references in the introduction with additional text outlining a brief review as suggested by the reviewer. Updated text on P1-2: “However, the risk of stranding also exists for low-carbon energy infrastructure in some contexts but has not been widely discussed [4, 5]. Investment in decentralised renewables-based off-grid solutions for energy access, such as solar photovoltaic (PV) mini-grids, has been inhibited by the risk of the arrival of national grid infrastructure as governments enact electricity-access plans [5–7]. Electricity tariffs from the national grid in developing countries like India and Ghana are significantly lower than those charged by mini-grids at cost recovery levels [8]. This is often due to subsidies or economies of scale. This could therefore compromise the ability of the decentralised infrastructure to provide a return on investment if competing side-by-side after the arrival of the national grid [9–11]. There is also a risk that mini-grid assets may be taken over by the government or distribution companies once the main grid arrives, potentially without proper compensation [7]. Consequently, there is a risk that the solar and battery assets installed to supply customers on the mini-grid may become stranded. Without clear policy support, such as designating zones for off-grid solutions, national grid-compatible technical standards, and commercial arrangements for generation and distribution assets [12], these risks limit the appetite of investors and companies for developing low-carbon mini-grid electricity infrastructure.”

#	Comment	Response
3	It is mentioned in the title as well as across the manuscript that one of the aims is to reduce emissions. The authors consider a linear decrease in emission intensity of grid-based electricity from a present value of 632 gCO₂eq/kWh to 440 gCO₂eq/kWh at the end of the eight-year modelling period. Supplementary Table 3 gives the numbers for the mini-grid. The sum of all GHGs – 1158 gCO₂eq/kW.	The difference here is that the values are not the same units. The 632 to 440gCO₂eq/kWh is the emissions intensity of the electricity produced. The 1158kgCO₂eq/kW (or kWp,, depending on the item) is the embedded emissions in the infrastructure due to its production, measured in terms of emissions per unit capacity.
4	Why is the unit of 'kWh' not used for mini-grid systems for the emissions intensity calculation?	Please refer to response 3 above. In Table 3 in the supplementary information when describing emissions, this is the embedded carbon in the infrastructure and is therefore given per unit capacity of infrastructure: kWp for solar panels, and kWh for battery storage.
5	Do the authors consider the complete life cycle emissions for the grid electricity as well?	We thank the reviewer for this comment and agree that considering the complete life-cycle emissions from national grids would help to determine their environmental impact more accurately. As the focus of our study is primarily on the economic viability of mini-grid systems, and the policy framework needed to mitigate the risk of asset stranding, the emissions analysis focuses primarily on whether providing hours of service beyond the national-grid network's reliability can offer both LCUE and emissions savings compared with the national-grid network. We do not consider the lifecycle emissions from the electricity supplied by the national grid, as doing so would require additional data and analysis beyond the scope of this study. Given this approach, the emissions intensity of the national grid, as currently presented, is underestimated. We have added text in the methods explaining this. On page 16: "We do not consider the full life cycle GHG emissions of the national grid electricity. As a result, our values of national grid emissions impact are underestimated."

#	Comment	Response
6	Why the consideration of linear decrease in emission intensity of grid-based electricity? What are the assumptions for this linear decrease?	For the carbon-emissions analysis conducted, we took two data points which could be verified by the literature: the current emissions intensity of 632 gCO₂eq/kWh and the emissions intensity that will result if India meets its NDC, namely, 440 gCO₂eq/kWh. Between these two values, with an uncertain political landscape (and a current target of net-zero by 2070), we have modelled the emissions intensity of the grid as a linear decrease. Linear decrease is consistent with other projections for emissions intensity evolution over time. However, there is a lot of uncertainty in terms of the electricity mix, but, in the short term, the value used is aligned with India's electricity plan for the next 10 years¹. Choosing an alternative future value, we believe, requires a careful analysis which would be beyond the scope of this manuscript. Similarly, as the primary focus of the manuscript is on the risk of asset stranding of decentralized renewable infrastructure, the emissions analysis is presented to give a representative picture of the relative carbon intensity of the mini-grid systems modelled; i.e., they are there to add to the picture of whether it is possible to both mitigate or avoid the risk of asset stranding whilst reducing carbon emissions. With an open-source methodology, as up-to-date data are published concerning India's national-grid carbon intensity, the conclusions drawn can be reframed should India exceed, or fail to meet, its NDCs. 1. https://cdnbbsr.s3waas.gov.in/s3716e1b8c6cd17b771da77391355749f3/uploads/2023/09/202309011256071349.pdf
7	Uttar Pradesh has close to 90% share of coal in its electricity generation. I assume the emission intensity will be higher than 632 gCO₂eq/kWh. Please check the following reference https://cea.nic.in/cdm-co2-baseline-database/?lang=en	We thank the reviewer for their insightful comment. We agree that the carbon intensity of production varies from region to region. In some contexts, power is generated and consumed at the state level with little cohesive policy between regions at a national level. We agree also that, from a production perspective, the share of coal in Uttar Pradesh is higher than the national average. However, in our study, we compare decentralised renewable infrastructure (namely, the PV-and-battery mini-grid) with a centralised national grid. In the case of a decentralised grid consumption occurs in the same region as production. In contrast, with a national grid,

#	Comment	Response
		electricity is traded between states based on demand. This is why we use the national grid's emission intensity instead of relying on production or plant-based emission intensity metrics. We have updated the text in the Methods on P17 to clarify this: “Although the electricity generation mix in Uttar Pradesh is particularly coal-heavy [76], we use the Indian national grid emissions intensity values in preference to values for Uttar Pradesh given that transmission of electricity occurs between different parts of India’s grid [77].”
8	Financial and technical assumptions: Do the authors consider changes in technical and financial parameters across the 8-year period? The assumptions in Supplementary Tables 2 and 3 are for which year? Please specify.	We do not consider changes in the cost or emissions values for the mini-grid infrastructure as we do not consider a ‘re-optimising’ of the system part way through its lifetime. All the infrastructure is installed in year zero, so changes in these values are not relevant. Cost data is from 2022 as stated in the caption to supplementary table 3. Other values, e.g. emissions data vary depending on data availability. We have added information regarding the years of publication of the sources used for the values provided in Supplementary Tables 2 and 3 in those tables of their captions.
9	Do the authors consider the influence of increasing standard of living till 2030 on appliance ownership and in general on the load profile? Please explain.	We thank the reviewer for their comment and agree that an increase in the standard of living is likely to correlate with a greater provision of electricity access and consumption. We also agree that, as electricity is provided to a community, loads are likely to increase. In the baseline scenarios modelled, we do not consider an increasing load: we look at the impact that a change in the reliability of the grid has on asset utilisation, and costs, over the lifetime of the mini-grid. We do explore the impact of a changing load on the LCUE and asset utilisation towards the end of the paper, with Figure 5 showing (in green) the impact that an increase in the load of 50% has on the LCUE and asset utilisation over the lifetime of the system. The data show that a 50% increase in demand results in only marginal increases in asset utilisation and a drop of <10% in the LCUE (from 0.21 \$/kWh to 0.19 \$/kWh). The shape of the demand curve we

#	Comment	Response
		have used will likely not be representative of real load growth within a community as there are multiple effects at play—increasing device ownership, population growth, and imitation, whereby consumers purchase devices inspired by neighbours etc. In this study, we have not aimed to predict the demand growth over time—though this has been done in the past utilising the modelling framework (namely, CLOVER) that we present [1]—and instead include load growth as a sensitivity analysis. 1. Sayani, R. et al. Sizing solar-based mini-grids for growing electricity demand: Insights from rural India. J. Phys. Energy 5, 014004 (2022).
10	Could the authors explain more why LCUE was used instead of LCOE? Shouldn't the cost of overgeneration (curtailment) be part of the cost calculation? If LCOE was used instead of LCUE, how would it affect the results?	For a mini-grid system powered by solar and storage, the proportion of electricity that is dumped and not used can be quite high. The costs of not utilising this electricity are implicitly accounted for: the LCUE takes the total costs incurred in providing electricity and divides it by the electricity that was consumed within the community. As such, any electricity that is dumped increases the LCUE as asset utilisation is decreased. Using a metric that includes the dumped electricity as well as the electricity consumed would distort the cost of electricity values downwards and would not be a fair comparison with the grid costs: the cost would be static, regardless of system size, and would not distinguish between oversized and optimised systems. In practical terms given the differences in the way these systems operate, LCUE for a mini-grid is the same as LCOE.
11	Abstract  • Please clarify if you are dealing with 'electricity' or 'electricity'. Line 1 gives reference to 'electricity', while line 2 refers to 'electricity' infrastructure. • 'We explore the impact of reduced demand to quantify...' Is it 'electricity' or 'electricity' demand. Please clarify to the readers. 	We thank the reviewer for this observation and agree there is a lack of clarity. We have corrected the abstract so that it is consistent for the reader, referring to electricity, and electricity demand in these instances mentioned.

#	Comment	Response
12	Introduction  Line 20: 'National grid electricity tariffs...' This statement is not true for all countries globally, consider the case of Australia as an example. There could be other countries as well. Check this publication https://www.sciencedirect.com/science/article/pii/S1364032121001581. Your references for the sentence are for India. So, please mention that the sentence is valid for a group of specific countries or India. 	Thanks for highlighting this point. We have revised the statement on page 2 in the introduction to clarify our point: “Electricity tariffs from the national grid in developing countries like India and Ghana are significantly lower than those charged by mini-grids at cost recovery levels [8]. This is often due to subsidies or economies of scale. This could therefore compromise the ability of the decentralised infrastructure to provide a return on investment if competing side-by-side after the arrival of national grid”
13	Methods  Line 377: The authors assume that '...an increase in the reliability of the national grid network would prompt customers to switch over from the mini-grid to using cheaper power sourced directly from the national grid'. What is the basis for such a strong assumption? 	We thank the reviewer for highlighting this point. It needs further clarification in the paper. This assumption rests on the fact that there is evidence that consumers will pay a premium for their electricity if they are getting a more reliable service [see https://www.sciencedirect.com/science/article/pii/S0973082617307615?via%3Dihub] for example. Therefore, if the national grid were to improve in its reliability, people would be increasingly less willing to pay the premium for the mini-grid electricity. In this paper we are examining a case to understand the transition risks posed by the arrival of the main grid in isolated areas where mini-grids are providing electricity or in underserved areas connected to the grid. We are not trying to predict how this would unfold in reality. Instead, we are showing the data from this scenario to explore the possibility, and to demonstrate the core point: a reduction in the number of consumers increases the risk of asset stranding. We have added a sentence in the text on page 10 where we explain the rationale for these scenarios: “These scenarios are based on the evidence that consumers are only willing to pay a premium for mini-grid electricity if it provides higher reliability [36].”

#	Comment	Response
14	The authors are assuming that grid-based electricity will still be cheaper than mini-grids even in 2030. Which might or may not be true. With the dramatic cost decline observed in the last decade for solar PV and Li-ion batteries, this might not hold true.	We agree with the reviewer that the costs of PV panels and batteries are decreasing with time, as is demonstrated across industry. In our study, we consider assets as installed once (i.e. , once a system is sized, no further renewable assets are installed). As such, we have taken present-day cost values and used these to assess the economic viability of the systems. We have not, in our study, considered the prospect of installing mini-grid systems at various intervals over coming years, wherein the CapEx costs for the renewable assets would be lower and, as such, developers would need to charge lower tariffs and be at less of a risk of asset stranding.
15	Line 87-88: Did you consider the revenue from selling excess electricity in the modelling? If so, what is the compensation for the sold electricity in \$/kWh?	We thank the reviewer for their comment. In our study, we consider the mini-grid as providing and selling electricity to the community, with a national grid network existing both as a source of competition and to provide power when the mini-grid is unable to meet demand. Whilst we agree that selling excess power generated could be a potential source of income for the mini-grid company, this is beyond the scope of our study where we chose to focus on the risk that changes in demand, and improvements in grid reliability, posed to the mini-grid in the context of asset stranding. We have added this point to the discussion, paragraph 4, page 13: “Alternatively, and beyond the scope of this study, facilitating mini-grids to also sell energy back to the grid could further improve operating revenues.” We are grateful to the reviewer for highlighting this potential future-work avenue.
16	Figure 3a,3b,3c: Please have the units on the left y-axis in brackets.	These have been added.

#	Comment	Response
17	Figure 2: The authors haven't mentioned about the assumption for kerosene lamps. Are they being used in any hour? Are the costs of purchasing fuel (kerosene) considered?	In our modelling, we do not consider kerosene lamps. We thank the reviewer for drawing this to our attention. We have moved any inadvertent references to kerosene from the manuscript.
18	Line 114-115: From Figure 2c, relying solely on the grid provides the cheapest LCUE – 0.16 \$/kWh. Please explain if the interpretation is wrong.	This interpretation is correct, but relies on the grid being available for 24 hours of the day. The data in Figure 2c. (and, more widely, across Figure 2) show the optimal systems when sized to meet a certain level of demand, given some availability of the national grid. E.G., in Figure 2c., the column with 24 hours of grid availability has equal costs throughout, because no assets (PV panels and batteries) need to be installed in order to supply electricity as this is already met by the national grid. In column 16, for instance, when 18, 20, 22 or 24 hours of service are required, the LCUE changes as mini-grid assets are installed to meet some of this demand, resulting in LCUE values of 0.2, 0.21, 0.23 and 0.28 \$/kWh respectively. Whilst the data in this figure do suggest that the grid can provide electricity most cheaply, not included here are the costs associated with this increase in reliability which may be passed onto the end consumer, i.e., a national grid which is available for 24 hours of the day is likely to cost more.

Reviewer 3

19	Key results - Improvement in the reliability of the national grid and subsequent longer hours of reliable electricity supply from the grid increases the levelized cost of used electricity. The tariff charged by mini-grid companies can cover the increase if the improvement is less than 50% and there are no critical times when both the national grid and the mini-grids cannot meet the demand.	We thank the reviewer for taking the time to read and review our paper. We hope that in addressing their comments, they feel that the paper has been strengthened.
----	--	---

20	2. Validity - The scope of stranded assets is less clear. Specify if it indicates only mini-grids (transmission and distribution systems) or includes both grids and decentralised solar generation. Given that the audience familiar with this research field knows the stranding of fossil fuel assets and the death spiral well, the precise definition is indispensable for accurate interpretation.	We thank the reviewer for this comment and we have added further clarification in the text to detail which assets we are referring to as being stranded: in our case, we are referring to the solar and storage infrastructure. New text in the intro (Page 2): “Consequently, there is a risk that the solar and battery assets installed to supply customers on the mini-grid may become stranded.”
21	The scenario of a 50% increase in demand with the same reliability looks like speculation and thus needs rationale or supporting evidence. The reviewer sees associated increases in demand with improvements in the national grid reliability make sense in the context of rural areas. There is plenty of evidence that an unreliable electricity supply has restricted purchases of electronic appliances such as fridges, cleaners, washing machines, and microwaves.	The situation specified by the reviewer is one scenario that has been considered in our study rather than a likely situation. In fact, our findings agree with the reviewer’s point and show that, in this scenario, increased demand results in a minimal increase in asset utilisation over the modelling period and only a small change in the average LCUE (from 0.21 to 0.19 \$/kWh, Figure 5). The 50% increase in demand is simply to illustrate the point that the installed assets are fairly well utilised and that demand increases alone cannot increase their utilisation. We include in the supplementary information (Supplementary Note 5) a figure (Supplementary Figure 5) exploring the impact of changes of demand on asset utilisation. Again, the changes in demand are purely illustrative: increases may be due to wider appliance ownership whilst decreases may be due to customers switching over to the national-grid connection. The results demonstrate the limited impact of an increase in demand on asset utilisation and the more significant impact on utilisation of decreases in demand. We agree that an unreliable electricity supply can stymie development as consumers put off purchasing appliances. However, the reliability in question is of the national-grid network rather than the overall electricity that is provisioned for. Again, we include in the supplementary information a figure (Supplementary Figure 6) exploring the relationship between reliability of the grid and asset utilisation to clarify this point.

22	3. Significance - The findings could provide a practical transition strategy for electricity systems in countries where the government prioritises rural electrification while incumbent grids are featured by coal lock-in.	We thank the reviewer for this comment and are glad they have made this judgment of the possible significance of the work. We hope that in addressing their detailed comments it will be further strengthened.
23	4. Data and methodology - Table 1 does not support the linearity between hours of operation and percentage changes in demand. The audience cannot understand why a 2- or 3-hour longer access to the national grid operation leads to incremental demand loss by 20% or 30%.	We agree with the reviewer that these relationships are unlikely to be linear: we don't know the true impact of changes in grid reliability on demand as there will be many socio-economic effects at play. The results presented are meant to be illustrative of potential scenarios, rather than accurate representation of reality, where we work on the basis that increased reliability will increase demand and vice-versa with no exact specification of this relationship given. We have clarified this point by including two new figures—Supplementary Figures 5 and 6—which decouple these two impacts and present results for changing grid reliability and user-base size separately in addition to the linked results presented (Figure 5).
24	Figure 4 shows 1-3 cents increase in LCUE with a demand loss of 20% or 50% instead of 1-2 cents in the main text (LL.182-3).	We thank the reviewer for drawing this to our attention. This has now been fixed, and the text in the manuscript matches the values in the figure and refers to 1-3 cents .
25	5. Analytical approach - The different referential points in the two scenario analyses devalue the results. The first analysis sets the reference point at 8 hours of grid availability (Figure 2&3), and the second at 13 hours (Figure 4&5). The inconsistency makes the results of the first analysis powerless and blocks comparisons of the impacts on carbon intensity between the two analyses.	We thank the reviewer for their comment and agree with them that, in its submitted form, the manuscript was laid out in a confusing manner. In Figures 2 and 3, the 8-hour reliability threshold is not a “reference” scenario; rather, it serves as the minimum level of service that we consider. The data presented in this section are for systems which are optimised to meet some level of service (8–24 hours of electricity, on average, per day) given some level of grid reliability. We consider only the range 8–23 hours of grid reliability and 8–24 of service provision in this section because a grid which is able to provide 24 hours of service would result in no mini-grid assets installed, whilst a grid which supplied less than 8 hours was deemed unrealistic given the current reliability of the national grid of 8.1 hours in Uttar Pradesh

		(reported in a 2019 study [1]), and the likeliness that this will increase over time. In Figures 4 and 5, we explore how changes in the grid reliability from the present-day value are likely to affect results. In this section, we are not exploring a range of hypothetical values for reliability and the optimum systems that would be installed, but, rather, are exploring how changes in reliability, over time, affect the viability of mini-grids. As such, we include a reference reliability of 13 hours, size assets to meet this demand, and explore how these installed assets perform under changes in reliability. 1. Dugoua, E., Kennedy, R., Shiran, M. & Urpelainen, J. Assessing reliability of electricity grid services from space: The437 case of Uttar Pradesh, India. Energy for Sustain. Dev. 68, 441–448, DOI: https://doi.org/10.1016/j.esd.2022.04.004 (2022).438
26	Use the same referential point in grid availability hours for the two scenario analyses.	We direct the reviewer to the above comment which we hope helps to clarify the approach taken in the manuscript.
27	Add a sensitivity analysis of the LCUE with mini-grids with and without imposing the cost of attaining India's NDC goal (531 gCO₂eq/kWh on average) for electricity imported from the national grid in the second analysis. Given that mini-grids coupled with decentralised solar generation are installed to increase electricity access without increasing carbon emissions, the comparison should be made on a level playing field. The manuscript would make a far more significant scholarly contribution to the journal Nature Communication. This improvement does not require new scenario analyses from scratch. It is sufficient to refer to the recent literature on the cost of carbon emissions, estimate it for the electricity imported from the national grid in India, and use the estimated cost for sensitivity analyses.	We agree with the authors that a sensitivity analysis implying the impact on the levelised cost of mini-grid systems of imposing a carbon tax on the electricity sourced from the national-grid network is of interest and would make a useful addition to the study. We have added Supplementary Note 6 which showcases the impact on the systems of the imposition of a carbon tax. Given the likely AR6 decarbonisation pathways [1], we imposed a carbon tax of 24.77 \$/tonne-CO₂eq emitted and used the carbon intensity shift required to meet India's NDC to compute the emissions intensity over time. We undertook optimisations which searched for systems which had the lowest LCUE given the minimum-service requirements imposed for a range of national-grid availabilities. The results demonstrated a minimal (less than 5%) impact on the LCUE and the data in Supplementary Figure 7 show at most a 16%

		shift away from the national-grid network. We attribute this to the fact that the majority of AR6 decarbonisation pathways demonstrate a minimal carbon price as the majority of the decarbonisation process takes place well after 2030. As such, local renewable assets are made significantly more cost-competitive only in cases where the grid was relied heavily upon; i.e., where limited additional service was required beyond that which the national-grid network could provide. In these cases, if the grid was not particularly reliable (and was supplying only a few hours of electricity per day), then installing some small additional number of local renewable assets can easily offset these costs. The impact of removing the grid subsidy (Supplementary Figure 3) shows the same trend in the LCUE but where 20% changes are visible (rather than 5% shifts) on account of the removal of the grid subsidy resulting in a much greater increase in cost than any imposition of a carbon tax.
28	Authors may discuss transmission and distribution development plans as a driver of improving the national grid reliability. An increasing number of emerging markets and developing economies have implemented plans recently to avoid unplanned installations of generation capacity and subsequent increases in grid development costs. For the case of the Philippines, see Delina et al., electricity Strategy Review 53, 2024, 101407.	We thank the reviewer for this comment. In response to your comment, we have added the following text to the discussion on P11-12: “Several countries, including India and Kenya, have launched initiatives such as the Green Energy Corridor Project and the Last-Mile Connectivity Project to enhance the reliability of their national grids and minimise the curtailment of renewable energy [64, 65]. Coordination between the government and private investors is essential, as understanding future national grid expansion and strengthening plans will allow mini-grid investors to effectively prepare for these changes and offer them more certainty over their ability to recoup investments alongside the national grid in the short and medium term”
29	The proposal of the exclusive right to supply a given set of customers within a zone designated for grid-connected mini-grids for a minimum length of time makes sense. Authors may refer to institutions of privately invested mini-grids in some countries and provide them as supporting evidence. While it becomes business as usual to conclude long-term power purchase agreements to attract private investment in renewable generation with storage and grid connection projects, the authors	We thank the reviewer for this comment. In response to your first point, we have added the following text to the discussion: “In countries like Nigeria and Tanzania, policies such as exclusive permits for developers and commercial agreements for generation and distribution assets—like tripartite agreements among distribution companies, communities, and operators— have enabled private

counter-argue this proposal in the paragraph starting from 241.	investment in this sector to provide reliable access to electricity in rural communities¹². We have modified the text in the discussion on P13 “A key consideration beyond the regulatory environment is that of the technical components (and costs) involved in mini-grid to national grid connection. National-grid connectivity adds complexity to mini-grid systems, as well as additional costs. Our study prioritises the use of lower-cost electricity from the national grid to assess the impact on the levelised cost (see Methods); however, in practice, control systems may be needed to determine under what conditions the mini-grid should prioritise the use of the national grid, local solar generation or stored electricity, depending on the power purchase agreement.”
--	--

Reviewer 1

30	The tension between centralized and decentralized modes of organizing the electricity generation capacity has been debated for several years. Its a proposition valid for both households who already have access and for those who don't. Decentralized generation has been explored to provision access through pilot installations, with much of the up-front cost being subsidized through grant aid. In absence of the grant aid the unit cost of generation are high. Such installations tend to suffer from lack of maintenance and reliability is often low. How or if such a model can be scaled has been a topic of much discussion. There are installations which recover their costs (plus profit) by charging tariffs which are higher than the grid as has been indicated in the surveyed literature. Usually, the last-mile connectivity is provided by the state utility, and the paying capacity of the Residents in these areas is low. Decentralized distributed generation for areas with access is also important component of the strategy to maximize deployment of renewable electricity. This strategy has its own challenges, including grid management, utility death spiral etc.	We would like to thank the reviewer for their insightful comments on the tension between centralised and decentralised electricity generation and its implications for energy access. We agree with the reviewer that there are challenges associated with decentralised electricity provision, including the ongoing maintenance, issues with reliability, and economic prospects of mini-grid systems, especially in the context of grant aid. Your points on the scalability of decentralised generation, along with the challenges related to tariff structures and the financial sustainability of such systems, are particularly valuable. The literature we reviewed does indeed highlight cases where decentralised systems have been able to recover costs but often at higher tariff rates compared to grid electricity. Regarding your comment on the role of decentralised distributed generation as part of a broader strategy for maximising renewable electricity deployment, we fully agree. The challenges around grid management and the potential risks to utility financial models, such as the "utility death spiral," are critical to consider.
-----------	--	--

31	In the framing of the problem, mini grids contribution to enhancing the affordability is included. This follows arguments and evidence on electricity usage enhancing the enterprise activity and hence the paying capacity, while also underscoring that mini-grids prices are higher than the grid, nonetheless people pay for it in absence of alternatives, indicating the value the customers place on reliable electricity. The claim in the following sections is of mini grids enhancing the affordability, and its not clear if its because of enhanced enterprise activity. In the modelling it becomes clear that this effect is not included and still the affordability increases across range of scenarios, which is counter-intuitive. In subsequent explanations, it comes out that this is on account of the additional cost of expanding the grid connections.	We thank the reviewer for their comment and their insight and we agree that so-called productive uses of energy are a viable means of reducing the costs of mini-grids [1]. We agree that it is initially counter-intuitive that renewable assets can provide power at a lower LCUE than using an existing national grid connection. However, this is primarily due to the counter-intuitive definition of 100% energy access purported by the Indian government [2], namely, that a village is “electrified” if 10% of households have access to electricity and public buildings (schools, clinics, etc.) have power [3]. In our demand modelling, we consider the demand of the entire community (all 547 households) and, as such, need to include the cost of extending the transmission network to reach all households. We agree that a potentially useful benchmark would be the current “10% of households” scenario. However, we believe that including this benchmark would be misleading: a 547-household mini-grid would have significantly more CapEx for providing electricity than the current 55-household scenario but is able to significantly improve livelihoods and supply power. We have updated the caption to Figure 1 to clarify this point as we agree with the reviewer that it was previously unclear in the paper: “Illustration showing the energy access scenarios compared in this paper: community without mini-grid infrastructure, where additional distribution infrastructure has been installed to extend access to all households in the community (left pane), and community with electricity access via national grid-connected mini-grid for enhanced reliability (right pane).”  1. Beath, H. et al. The cost and emissions advantages of incorporating anchor loads into solar mini-grids in India. Renewable and Sustainable Energy Transition 1, 100003 (2021). 2. D’Cunha, S. D. Modi Announces ‘100% Village Electrification’, But 31 Million Indian Homes Are Still In The Dark. Forbes Magazine (2018). 3. Agrawal, A., Kumar, A. & Joji Rao, T. 100% Rural Electrification in India: Myth or Reality? in Energy, Environment and Globalization: Recent Trends, Opportunities and Challenges in India (eds. Gupta, A. & Dalei, N. N.)

		117–126 (Springer Singapore, Singapore, 2020). doi:10.1007/978-981-13-9310-5_6
32	Page 6, line 101-7: For national grid the additional cost of distribution network for the grid is socialized over larger network. Recovering this cost from the electricity sold to the local community is not the practice. This seems to be main driver of the results, though it is not discussed sufficiently and justified.	We agree with the reviewer that the general practice when carrying out rural electrification through the extension of the national grid network is not to recover the costs associated with the transmission and distribution infrastructure from the local community. The data we present are not tariffs which are charged to consumers. The results we present in this section (shown in Figure 2) are the levelised cost of used electricity (LCUE): the total costs incurred when providing electricity divided by the total electricity that was consumed. Whilst the LCUE can be related to tariffs charged, with mini-grid companies aiming to recover their costs through consumers, they cannot be directly equated. However, the LCUE can be useful in informing policy-makers and mini-grid installers as to the most cost-effective means of providing electricity access. We have added the following text to the manuscript: “It should be noted that these values for the LCUE refer to the costs incurred per levelised cost of providing electricity which is consumed. Not included in this metric is the actual cost which would be charged to a consumer, where mini-grid companies would likely seek to recover costs incurred and generate a profit over the lifetime of the system. Rather than incorporate an estimate of likely profit margins, we here use the widely utilised metric of LCUE to enable a good comparison between our systems and alternative means of supplying electricity.”
33	Separately, have authors come across reliability reports for mini grids? I have come across some reports of neglect and lack of upkeep for a few such mini grid installations.	We thank the reviewer for this point and agree that it is important to raise. Regarding the reliability of mini-grids, there is evidence of both neglect (and poor reliability), but also evidence that mini-grid electricity access can provide improved reliability. We have added some text and additional references in the introduction, highlighting that better reliability is conditional on effective maintenance.

		Updated text on P2 of the introduction: “Mini-grids have the potential to offer consumers more reliable access to electricity compared to national grids [13, 14], provided they are well-maintained and protected from severe weather [15].”
34	Figure 2b) The legend should be labelled battery capacity.	We thank the reviewer for drawing this to our attention; this has now been fixed.
35	In the results section: “Regardless of the national grid availability level (in hours of service provided), increasing the minimum service hours by similar amounts (e.g., by two, from four to six hours per day) requires similar increases in PV capacity.” The increase in PV capacity qualified by “similar”. Could the authors clarify the similarity? PV capacity increases for increasing service hours, the quantum of increase is different. The increase is not proportional either.	We agree with the reviewer that this point was confusing and could not be well quantified. We have removed this comment from the manuscript.
36	Page 6, line 101-7: 110-12: Grid emission intensity will change with increasing centralized solar and wind installations.	Yes, the reviewer is right that the grid emissions intensity of electricity will reduce as the proportion of renewable energy increases on the Indian grid. It is for this reason that we assume a reduction in the grid emissions intensity over time. We have made this clearer in the results text where we discuss emissions on page 6: “The average emissions intensity of the electricity used (Figure 2d) for mini-grid systems with PV and storage in the region is below India’s NDC goal (531 gCO₂eq/kWh on average; although we assume a reduction over time due to increased renewables¹⁰⁹ penetration, see Methods)...” In the Methods, we state the following on page 17: “The emissions intensity of grid electricity linearly decreases from a present value of 632 gCO₂eq/kWh to 440 gCO₂eq/kWh at the end of the eight-year modelling period”

Response to Reviewers: NCOMMS-24-28162-T

Reviewer 2

#	Comment	Response
1	Thank you for revising the manuscript. The manuscript has been revised with appropriate responses and text additions in the article. I do not have any further comments.	We thank the reviewer for their time considering our manuscript and are pleased they feel that we have addressed their comments.

Reviewer 3

0.1	The manuscript does not present past research and research gaps, which makes it difficult for the reviewer to evaluate originality and novelty.	We thank the reviewer for this comment. In our introduction, whilst we aim to give a broad overview and adequate context for non-experts, we present previous relevant research specifically on national grid-connected mini-grids. On page page 3 lines 64–66: “Previous studies have used a single reliability level for the national grid network (52–55). However, it is important to explore the impacts at different national grid reliability levels, given that it may differ from what is anticipated, or change over time” We are confident in the novelty this work brings as it explores many more scenarios of grid reliability and considers the impact of lost demand on the LCUE. In this study, we also assess the risks to the decentralised systems across various scenarios; these haven’t been quantified in the previous studies.
0.2	Provide assumptions on the villagers' electricity access when inaccessible to national grid electricity. Do they use charcoal or other biomass-based fuel as alternatives, or live without any electricity services? Do they reduce consumption of these alternatives but keep using them unless they can access electricity for 24 hours with the national grid and local mini-grid combined? The assumption may affect the LCUE and emission intensity.	We thank the reviewer for their comment. Whilst we agree that this is an important thing to consider, and whilst our model (CLOVER) does allow for the inclusion of kerosene—the alternative to electric lighting—we do not include these impacts due to the complexities and uncertainties in doing so; namely, the additional level of uncertainty around the alternative lighting sources employed—whether solar lanterns, home-made SHS or kerosene lamps—and how these will change over time. This is a shortcoming of our study, and we have now updated our methods section to include this limitation.

		Had we included kerosene usage, it would have... and, thus, we have effectively underestimated the emissions and cost benefits of switching to the mini-grid system. See page 16: “Our analysis does not account for the use of kerosene as an alternative to electric lighting, despite the CLOVER model having the functionality to include it. This is due to complexities and uncertainties in estimating its use and impacts. This omission represents a limitation of our study. If kerosene were to be included, it would be likely to further improve the emissions savings of switching to the mini-grid over all hours of grid availability and service provision.”
1	Figures 2c and 2d and their explanation should be placed after the next section elaborating on LCUE. They are too complicated and demanding for the audience to understand the results only with these figures and explanations.	We thank the reviewer for their comment and agree with their assessment that Figure 2 is best placed after the next section. We have moved Figure 2 in the manuscript to appear after the next section. Further, as the manuscript will change in format at the typesetting stage, we will make this known to the type-setting team.
2	Rephrase the sentence in Lines 115-116 “with only mini-grid systems providing 20 or more hours of service with a national grid availability of 18 or more hours per day exceeding this target” in plain English.	We agree with the reviewer that the original phrasing of this was unclear. We have now rephrased this, and the text in the manuscript reads as follows: “where only systems which are sized for high levels of service (20 or more hours per day) when the national-grid network is readily available (for 18 or more hours per day on average) are more carbon intensive.”
3	The paragraph starting from Line 204 may be replaced with research questions. It looks like a repetition of the motivations in the Introduction. Reminding research questions is much more productive.	We thank the reviewer for this comment and agree that there was too much repetition here. We have rewritten and restructured this paragraph so that the recap of the problem statement is much shorter, and we then summarise the research questions that the paper addresses, and our findings in response to them (pages 11–12): “National grid networks provide chronically unreliable electricity for many rural users in low- and middle-income countries [16], including India [5,33,34] where a significant proportion of the population remain unserved [30,64]. Supply can be unpredictable, and, in many cases, is also highly carbon-intensive. Whilst the established strategy for providing access to electricity is to strengthen and expand national

		grids, this can be expensive on a per-connection basis in lower-density rural areas that are further from urban centres [16]. In response to these challenges, we set out to address the following research questions:  1) To what extent can national grid-connected solar PV mini-grids improve electricity service hours for consumers across different reliability scenarios, and what are the costs and emissions of this approach? 2) How do future potential changes in national-grid reliability and consumer demand affect the risk of asset stranding and the long-term financial viability and environmental sustainability of solar PV mini-grids? 3) What policies can be implemented in order to better mitigate against the risk of asset stranding and better effect the interconnection of mini-grids with the national grid? We find that local renewable mini-grid networks may be able to boost hours of supply for rural users whilst requiring lower investments than national-grid extension. We find that doing so can also simultaneously reduce emissions across a range of grid reliabilities and levels of service provision. We find that there is a risk of asset stranding when the reliability of the national-grid network improves. When quantified, however, the increase in the LCUE is minimal compared to the tariffs commonly charged by mini-grid developers.”
4	The paragraph starting from line 217 is redundant. It must be consolidated with the introduction and the policy implications starting from line 237. It is odd to see similar policy implications separately.	We thank the reviewer for their comment. The three paragraphs on policy, beginning with paragraph beginning on line 217, introduce policy implications from our study which we believe are worth highlighting in order for our study to effect the greatest change. As such, we are keen to include a rich and full discussion of the policy implications of our findings. We agree with the reviewer, however, that the structure of this discussion did not best serve the content and have effected a restructuring of the section—reworking the content contained in the various paragraphs—which we hope highlights the policy discussion better.

5	Lines 229–231 do not make sense from an economic viewpoint. The interpretation may be improved if rephrasing: 1) demand loss from an improved grid substantially reduces revenue for mini-grids, generating stranded costs. 2) the demand loss increases the LCUE only slightly, thus does not generate further demand loss and increases in stranded costs.	We thank the reviewer for their comment. On lines 229–231, we discuss the impact on the LCUE of simultaneously improving the reliability of the national-grid supply and reducing the demand as a proxy for customer-base reduction as consumers switch over to using the national-grid network. The reviewer is right to point out the potential link between an increase in the LCUE and further demand losses, though the exact mechanism for this depends on whether developers absorb the cost increases or pass these onto the consumer by charging higher tariffs. As the scenarios we modelled are designed to showcase potential couplings between reliability improvements and demand loss, the exact link between these two is beyond the scope of our study. However, we have added text to the manuscript on page 19 to point out this potential link as a limitation and an avenue for future work: “Potential second-order effects, whereby the increase in the LCUE resulting from a decrease in asset utilisation results in further demand loss, is not considered here. Additional scenarios with varying degrees of demand loss for fixed levels of grid reliability are outlined in the Supplementary Information.”
6	Policy implication to India in Lines 264–266 is insufficient. The results reveal that perceived stranded risks make private investors hesitant to invest in local mini-grids. Discuss and suggest policies that can reduce the risks and their perception.	We thank the reviewer for their comment and for highlighting the importance of policy in our manuscript. We have discussed a range of policy implications in our manuscript. These include technical standards, integration policies, and creating an environment to protect mini-grid investors. We have discussed a range of policies which are relevant not just to India, but also to countries that have similar challenges more broadly. The reviewer rightly points out that our paper raises the issue of stranding risk. We have included in our manuscript detail on policies that could be implemented in India, or in other contexts, to protect investors. This paragraph is now lines 239–251 on pages 12–13. In this paragraph, we present policy recommendations such as exclusive operating zones, which are less tied to the context of India and, as such, are applicable more broadly.

7	Methodology is sound. However, the session title of “Scenarios including electricity demand loss and impacts on stranded asset risk” in Line 179 may be replaced with “sensitivity analysis of demand and LCUE” to represent the subsection.	We thank the reviewer for their suggestion and agree that the previous section title was not clear enough. We have taken on-board their suggestion and have incorporated it, retitling the section: “Sensitivity analysis of changes in demand on LCUE and the stranded asset risk”
---	---	--